

# The atmospheric impacts of monoterpene ozonolysis on global stabilised Criegee intermediate budgets and SO₂ oxidation: experiment, theory and modelling

**Mike J. Newland[1,3], Andrew R. Rickard[2,3], Tomás Sherwen[3], Mathew J. Evans[2,3], Luc Vereecken[4,5], Amalia Muñoz[6], Milagros Ródenas[6], William J. Bloss[1]**

[1]{University of Birmingham, School of Geography, Earth and Environmental Sciences, Birmingham, UK}

[2]{National Centre for Atmospheric Science (NCAS), University of York, York, UK}

[3]{Wolfson Atmospheric Chemistry Laboratories, Department of Chemistry, University of York, York, UK}

[4]{Max Planck Institute for Chemistry, Atmospheric Sciences, Hahn-Meitner-Weg 1, Mainz, Germany}

[5]{Institute for Energy and Climate Research, Forschungszentrum Jülich GmbH, Jülich, Germany}

[6]{Fundación CEAM, EUPHORE Laboratories, Avda/Charles R. Darwin 14. Parque Tecnologico, Valencia, Spain}

Correspondence to:   M. J. Newland (mike.newland@york.ac.uk)

A. R. Rickard (andrew.rickard@york.ac.uk)

## Abstract

The gas-phase reaction of alkenes with ozone is known to produce stabilised Criegee intermediates (SCIs). These biradical/zwitterionic species have the potential to act as atmospheric oxidants for trace pollutants such as SO₂, enhancing the formation of sulfate aerosol with impacts on air quality and health, radiative transfer and climate. However, the importance of this chemistry is uncertain as a consequence of limited understanding of the abundance and atmospheric fate of SCIs. In this work we apply experimental, theoretical and numerical modelling methods to quantify the atmospheric impacts, abundance, and fate, of



the structurally diverse SCIs derived from the ozonolysis of monoterpenes, the second most
abundant group of unsaturated hydrocarbons in the atmosphere. We have investigated the
removal of $SO_2$ by SCI formed from the ozonolysis of three monoterpenes (α-pinene, β-
pinene and limonene) in the presence of varying amounts of water vapour in large-scale
simulation chamber experiments. The $SO_2$ removal displays a clear dependence on water
vapour concentration, but this dependence is not linear across the range of $[H_2O]$ explored. At
low $[H_2O]$ a strong dependence of $SO_2$ removal on $[H_2O]$ is observed, while at higher $[H_2O]$
this dependence becomes much weaker. This is interpreted as being caused by the production
of a variety of structurally (and hence chemically) different SCI in each of the systems
studied, each displaying different rates of reaction with water and of unimolecular
rearrangement/decomposition. The determined rate constants, $k(SCI+H_2O)$, for those SCI that
react primarily with $H_2O$ range from $4 - 310 \times 10^{-15}$ cm$^3$ s$^{-1}$. For those SCI that predominantly
react unimolecularly, determined rates range from $130 - 240$ s$^{-1}$. These values are in line with
previous results for the (analogous) stereo-specific SCI system of *syn/anti*-$CH_3CHOO$. The
experimental results are interpreted through theoretical studies of the SCI unimolecular
reactions and bimolecular reactions with $H_2O$, characterised for α-pinene and β-pinene at the
M06-2X/aug-cc-pVTZ level of theory. The theoretically derived rates agree with the
experimental results within the uncertainties. A global modelling study, applying the
experimental results within the GEOS-Chem chemical transport model, suggests that > 98 %
of the total monoterpene derived global SCI burden is comprised of SCI whose structure
determines that they react slowly with water, and whose atmospheric fate is dominated by
unimolecular reactions. Seasonally averaged boundary layer concentrations of monoterpene-
derived SCI reach up to $1.2 \times 10^4$ cm$^{-3}$ in regions of elevated monoterpene emissions in the
tropics. Reactions of monoterpene derived SCI with $SO_2$ account for < 1 % globally but may
account for up to 50 % of the gas-phase $SO_2$ removal over areas of tropical forests, with
significant localised impacts on the formation of sulfate aerosol, and hence the lifetime and
distribution of $SO_2$.
**1   Introduction**
Chemical oxidation processes in the atmosphere exert a major influence on atmospheric
composition, leading to the removal of primary emitted species, and the formation of
secondary products. In many cases either the emitted species or their oxidation products





negatively impact air quality and climate (e.g. ozone, which is also a potent greenhouse gas).
These reactions can also transform gas-phase species to the condensed phase, forming
secondary aerosol that again can be harmful to health and can both directly and indirectly
influence radiative transfer and hence climate (e.g. $SO_2$ oxidation leading to the formation of
sulfate aerosol).
Tropospheric gas-phase oxidants include the OH radical, ozone, the $NO_3$ radical, and halogen
atoms. Stabilised Criegee intermediates (SCIs), or carbonyl oxides, have been identified
as another potentially important oxidant in the troposphere (*e.g.* Cox and Penkett, 1971;
Mauldin et al., 2012). SCIs are thought to be formed in the atmosphere predominantly
from the reaction of ozone with unsaturated hydrocarbons, though other processes may
be important under certain conditions, e.g. alkyl iodide photolysis (Gravestock et al.,
2010), dissociation of the DMSO peroxy radical (Asatryan and Bozzelli, 2008).
Laboratory experiments and theoretical calculations have shown SCI to oxidise $SO_2$ (*e.g.*
Cox and Penkett, 1971; Welz et al., 2012; Taatjes et al., 2013), organic (Welz et al.,
2014) and inorganic (Foreman et al., 2016) acids (Vereecken et al., 2017), and a number
of other important trace gases found in the atmosphere, as well as forming adducts with
$NO_2$ (Taatjes et al., 2014; Vereecken et al., 2017; Caravan et al., 2017). Measurements in
a boreal forest (Mauldin et al., 2012) and at a coastal site (Berresheim et al., 2014) have
both identified a 'missing' process (in addition to reaction with OH) oxidising $SO_2$ to
$H_2SO_4$, potentially arising from SCI reactions.
Here, we present results from a series of experimental studies into SCI formation and
reactions, carried out under atmospheric boundary layer conditions in the European
Photochemical Reactor facility (EUPHORE), Valencia, Spain. We examine the ozonolysis of
three monoterpenes with very different structures (and hence reactivities with OH and ozone):
α-pinene (with an endocyclic double bond), β-pinene (with an exocyclic double bond) and
limonene (with both an endo and exo cyclic double bond). We observe the removal of $SO_2$ in
the presence of each alkene-ozone system as a function of water vapour concentration. This
allows us to derive relative SCI kinetics for reaction with $H_2O$, $SO_2$, and unimolecular
decomposition. Further, we calculate absolute unimolecular rates and bimolecular reaction
rates with $H_2O$ for all α-pinene and β-pinene derived SCI at the M06-2X/aug-cc-pVTZ level
of theory. A global modelling study, using the GEOS-Chem global chemical transport model,



is performed to assess global and regional impacts of the chemical kinetics of monoterpene
SCI determined in this study.

## 1.1  Stabilised Criegee Intermediate Kinetics

Ozonolysis of an unsaturated hydrocarbon produces a primary ozonide that rapidly
decomposes to yield pairs of Criegee intermediates (CIs) and carbonyls (Johnson and
Marston 2008). The population of CIs are formed with a broad internal energy
distribution giving both chemically activated and stabilised forms. Chemically activated
CIs may undergo collisional stabilisation to an SCI, unimolecular decomposition or
isomerisation. SCIs can have sufficiently long lifetimes to undergo bimolecular reactions
(Scheme 1).
The predominant atmospheric fate for the simplest SCI, $CH_2OO$, is reaction with water
vapour (likely with the dimer ($(H_2O)_2$) (e.g. Berndt et al., 2014; Newland et al., 2015a;
Chao et al., 2015; Lewis et al., 2015; Lin et al., 2016). For larger SCI, both experimental
(Taatjes et al., 2013; Sheps et al., 2014; Newland et al., 2015a; Huang et al., 2015) and
theoretical (Kuwata et al., 2010; Anglada et al., 2011; Anglada and Sole, 2016) studies
have shown that their kinetics, in particular reaction with water, are highly structure
dependent. The significant double bond character exhibited in the zwitterionic
configurations of mono-substituted SCI leads to two distinct chemical forms: *syn*-SCI
(*i.e.* those where an alkyl-substituent group is on the same side as the terminal oxygen of
the carbonyl oxide moiety)), and *anti*-SCI (*i.e.* with the terminal oxygen of the carbonyl
oxide moiety on the same side as a hydrogen group). The two conformers of $CH_3CHOO$,
which are both mono-substituted, display these properties. This difference in conformer
reactivities has been predicted theoretically (Ryzhkov and Ariya, 2004, Kuwata et al.,
2010; Anglada et al., 2011; Lin et al., 2016) and was subsequently confirmed
experimentally (Taatjes et al., 2013; Sheps et al., 2014) for the two $CH_3CHOO$
conformers. The significantly faster reaction of *anti*-$CH_3CHOO$ with water is driven by
the higher potential energy of this isomer, while more stable SCI, with a methyl group in
*syn*-position, such as *syn*-$CH_3CHOO$ or $(CH_3)_2COO$, react orders of magnitude more
slowly with water.
SCI can also undergo unimolecular isomerisation/decomposition in competition with
bimolecular reactions. This is likely to be a significant atmospheric sink for *syn*-SCI because



of their slow reaction with water vapour (e.g. Huang et al., 2015). Unimolecular reactions of
*syn*-CI/SCI are dominated by a 1,4-H-shift, forming a vinyl hydroperoxide (VHP)
intermediate (Niki et al., 1987; Rickard et al., 1999; Martinez and Herron, 1987; Johnson and
Marston, 2008; Kidwell et al., 2016). Decomposition of the VHP formed in this process is an
important non-photolytic source of OH, $HO_2$, and $RO_2$ in the atmosphere (Niki et al.,
1987; Alam et al., 2013; Kidwell et al., 2016), which can also lead to secondary organic
aerosol formation (Ehn et al., 2014). Unimolecular reactions of the *anti*-CI/SCI are
thought to be dominated by a 1,3-ring closure, the "acid/ester channel", in which the
CI/SCI decomposes, through a rearrangement to a dioxirane intermediate, producing a
range of daughter products and contributing to the observed overall $HO_x$ radical yield
(Kroll et al., 2002; Johnson and Marston, 2008; Alam et al., 2013).
$$\text{Alkene} + O_3 \xrightarrow{k_1} \phi\text{SCI} + (1 - \phi)\text{CI} + \text{RCHO} \tag{R1}$$
$$\text{SCI} + SO_2 \xrightarrow{k_2} SO_3 + \text{RCHO} \tag{R2}$$
$$\text{SCI} + H_2O \xrightarrow{k_3} Products \tag{R3}$$
$$\text{SCI} \xrightarrow{k_d} Products \tag{R4}$$
$$\text{SCI} + acid \xrightarrow{k_5} Products \tag{R5}$$
$$\text{SCI} + (H_2O)_2 \xrightarrow{k_6} Products \tag{R6}$$
Decomposition of the simplest SCI, $CH_2OO$, is slow ($< 10$ $s^{-1}$) and is not likely to be an
important sink in the troposphere (e.g. Newland et al., 2015a; Chhantyal-Pun et al., 2015).
This decomposition occurs primarily via rearrangement through a 'hot' acid species, which
represents the lowest accessible decomposition channel (Gutbrod et al., 1996; Alam et al.,
2011; Chen et al., 2016). However, recently determined unimolecular reaction rates of larger
*syn*-SCI are considerably faster. Newland et al. (2015a) reported unimolecular reaction rate
constants for *syn*-$CH_3CHOO$ of 348 ($\pm$ 332) $s^{-1}$ and for $(CH_3)_2COO$ of 819 ($\pm$ 190) $s^{-1}$
(assuming $k(syn\text{-}CH_3CHOO + SO_2)$ = 2.9 $\times$ $10^{-11}$ $cm^3$ $s^{-1}$ (Sheps et al., 2014) and
$k((CH_3)_2COO + SO_2)$ = 1.3 $\times$ $10^{-10}$ $cm^3$ $s^{-1}$ (Huang et al., 2015), respectively). Smith et al.
(2016) measured the unimolecular decomposition rate of $(CH_3)_2COO$ to be 269 ($\pm$ 82) $s^{-1}$ at
283 K increasing to 916 ($\pm$ 56) $s^{-1}$ at 323 K, suggesting the rate to be fast and highly
temperature dependent. Novelli et al. (2014), estimated a significantly slower decomposition





rate for s*yn*-CH$_3$CHOO of 20 (3-30) s$^{-1}$ from direct observations of OH formation, while
Fenske et al. (2000), estimated the decomposition rate of CH$_3$CHOO (i.e. a mix of *syn* and
*anti* conformers) produced from ozonolysis of *trans*-but-2-ene to be 76 s$^{-1}$ (accurate to within
a factor of three).
## 1.2 Monoterpene Ozonolysis
Monoterpenes are volatile organic compounds (VOCs) with the general formula C$_{10}$H$_{16}$,
which are emitted by a wide range of vegetation, particularly from boreal forests. Total global
monoterpene emissions are estimated to be 95 (± 3) Tg yr$^{-1}$ (Sindelarova et al., 2014) -
roughly 13 % of total non-methane biogenic VOC emissions. Monoterpene emissions are
dominated by α-pinene, which accounts for roughly 34 % of the total global emissions, while
β-pinene and limonene account for 17 % and 9 % respectively (Sindelarova et al., 2014).
Monoterpenes (mainly α-pinene and limonene) are also present in indoor environments, in
significant amounts where cleaning products and air fresheners are in routine use (on the
order of 100s of ppbv) (e.g. Singer et al., (2006); Sarwar and Corsi, (2007)), where their
ozonolysis products can affect indoor chemistry and health (*e.g.* Rossignol et al., (2013);
Shallcross et al., (2014)).
Monoterpenes are highly reactive due to the presence of (often multiple) double bonds. The
oxidation of monoterpenes yields a wide range of multi-functional gas-phase and aerosol
products. This process can be initiated by OH and NO$_3$ radicals or by O$_3$, with ozonolysis
having been shown to be particularly efficient at generating low volatility products that can
form SOA, even in the absence of sulfuric acid (e.g. Ehn et al., 2014; Kirkby et al., 2016).
These highly oxygenated secondary products have received considerable attention in recent
years because of their role in affecting the climate through absorption and scattering of solar
radiation (the direct aerosol effect). They can also increase cloud condensation nuclei
concentrations, which can change cloud properties and lifetimes (the indirect aerosol effect).
They have also been shown to have a wide range of deleterious effects on human health (e.g.
Pöschl and Shiraiwa, 2015).
The ozonolysis reaction for monoterpenes is expected to follow a similar initial process to
that of smaller alkenes, with cyclo-addition at a double bond giving a primary ozonide (POZ),
followed by rapid decomposition of the POZ to yield a CI and a carbonyl (Scheme 1).





Stabilisation of the large POZs formed in monoterpene ozonolysis is expected to be
negligible (Nguyen et al., 2009). However, a major difference in ozonolysis at endocyclic
bonds is that, on decomposition of the POZ, the carbonyl oxide and carbonyl moieties are
tethered as part of the same molecule, providing the potential for further interaction of the
two. These can react together to form secondary ozonides (SOZ), which may be stable for
several hours (Beck et al., 2011). However, while this has been shown to be potentially the
major fate in the atmosphere for SCI derived from sesquiterpenes ($C_{15}H_{24}$) (e.g. Nguyen et al.,
2009b; Beck et al., 2011; Yao et al., 2014), formation of SOZ is predicted to be small for
monoterpene derived SCI because of the high ring strain caused by the tight cyclisation (e.g.
Nguyen et al., 2009b). Chuong et al. (2004) predicted formation of a SOZ to become the
dominant atmospheric fate for SCI formed in the ozonolysis of endo-cyclic alkenes with a
carbon number between 8 and 15, while Vereecken and Francisco (2012) suggested that
internal SOZ formation is likely to be limited to product rings containing six or more carbons
due to ring strain.
No studies have yet directly determined the reaction rates of the large SCI produced from
monoterpene ozonolysis with $SO_2$ (or any other trace gases). This is owing to the
complexities of synthesizing and measuring large SCI. However, Ahrens et al. (2014)
concluded that the reaction of the C9-SCI formed in β-pinene ozonolysis with $SO_2$ is as fast
as that determined by Welz et al. (2012) and Taatjes et al. (2013) for $CH_2OO$ and $CH_3CHOO$
respectively (ca. $4 \times 10^{-11}$ $cm^3$ $s^{-1}$) by fitting to the decay of $SO_2$ in the presence of the
ozonolysis reaction. Mauldin et al. (2012) calculated significantly slower reaction rates for an
additional oxidant (assumed to be SCI) derived from α-pinene and limonene ozonolysis, with
$k$(SCI+$SO_2$) determined to be $6 \times 10^{-13}$ $cm^3$ $s^{-1}$ and $8 \times 10^{-13}$ $cm^3$ $s^{-1}$ for α-pinene and limonene
derived SCI respectively. However, it seems likely that the rates calculated by Mauldin et al.
(2012) may be substantially underestimated due to the assumption of a very long SCI lifetime
(0.2 s) in experiments that were performed at 50 % RH. The calculated rates scale linearly
with SCI lifetime and based on reaction rates of smaller SCI with $H_2O$ (reported since the
Mauldin et al. work, e.g. Taatjes et al., 2013) it seems likely that the lifetime of the SCI in
their experiments would have been more like $0.1 – 2 \times 10^{-2}$ s, increasing the calculated rate
constants by more than an order of magnitude, bringing them into much closer agreement
with the rates reported by Ahrens et al. (2014).
Unimolecular reactions of the monoterpene SCI are expected to proceed rapidly through the
VHP route if a hydrogen is available for a 1,4 H-shift. Those SCI that cannot undergo this





rearrangement may undergo unimolecular reactions via the formation of the dioxirane
intermediate, but this is expected to be a much slower process (Nguyen et al., 2009). In
contrast to smaller SCI, it has been observed experimentally, and predicted theoretically, that
the VHP route will mainly lead to rearrangement to an acid (also yielding an OH radical)
rather than decomposition of the molecule (e.g. Ma et al., 2008, Ma and Marston, 2008). As
for the smaller alkenes, monoterpene ozonolysis has been shown to be a source of $HO_x$ (e.g.
Paulson et al., 1997; Alam et al., 2013), predominantly via the VHP rearrangement. The
MCMv3.3.1 (Jenkin et al., 2015) applies OH yields of 0.80, 0.35 and 0.87 for α-pinene, β-
pinene and limonene respectively.

### 1.2.1  α-pinene derived SCI

Decomposition of the α-pinene POZ yields four different $C_{10}$ Criegee intermediates (Scheme
2: CI-1a, 1b, 2a, 2b), with the carbonyl oxide moiety at one end and a carbonyl group at the
other. Here, CI-1 is a mono-substituted CI for which both *syn* (1a) and *anti* (1b) conformers
exist, while the other, CI-2, is di-substituted, for which two *syn*-conformers (2a and 2b) exist.
Ma et al. (2008) infer a relative yield of 50 % for the two basic CI formed, based on the
observation that norpinonic acid yields from the ozonolysis of α-pinene and an enone, which
upon ozonolysis yields CI-1, are almost indistinguishable.
The total SCI yield from α-pinene was determined to be 0.15 (± 0.07) by Sipilä et al. (2014)
in indirect experiments measuring the production of $H_2SO_4$ from $SO_2$ oxidation in the α-
pinene ozonolysis system. Drozd and Donahue (2011) also determined a total SCI yield of
about 0.15 at 740 Torr, from measuring the loss of hydrofluoroacetone in ozonolysis
experiments in a high pressure flow system. The MCMv3.3.1 (Jenkin et al., 1997; Saunders et
al., 2003; Jenkin et al., 2015) applies a value of 0.20 based on stabilisation of only the mono-
substituted CI-1.

### 1.2.2  β-pinene derived SCI

β-pinene ozonolysis yields two distinct conformers of the nopinone C9-CI (Scheme 3: CI-3
and CI-4), which differ in orientation of the carbonyl oxide group, and $CH_2OO$. CI-3 and CI-4
are formed in roughly equal proportions with very little inter-conversion between the two
(Nguyen et al., 2009). The difference in the chemical behaviour of CI-3 and CI-4, which were
often not distinguished in earlier studies, arises from the inability of the carbon attached to the
four-membered ring to undergo the 1,4-H-shift that allows unimolecular decomposition via





the VHP channel. This was noted in Rickard et al. (1999) as being a reason for the
considerably lower OH yield (obtained via the VHP route) from β-pinene ozonolysis
compared to that of α-pinene. This difference leads to contrasting unimolecular
decomposition rates for the two CI, with Nguyen et al. (2009) predicting a loss rate of *ca.* 50
s$^{-1}$ for CI-3 (via a VHP) and *ca.* 1 s$^{-1}$ for CI-4 (via ring closure to a dioxirane). This result is
qualitatively consistent with the experimental work of Ahrens et al. (2014), who determine a
ratio of 85:15 for the abundance of SCI-4:SCI-3 about 10 s after the initiation of the
ozonolysis reaction, as a consequence of the much faster decomposition rate of SCI-3. Thus
the potential for bimolecular reactions to compete with decomposition of SCI-3 and SCI-4 in
the atmosphere is very different.
Nguyen et al. (2009) theoretically calculate a total SCI yield from β-pinene ozonolysis of 42
%, consisting of 16.2 % SCI-3, 20.6 % SCI-4, and 5.1 % $CH_2OO$. Ahrens et al. (2014)
assume an equal yield of CI-3 and CI-4 (45 %) with a 10 % yield of $CH_2OO$; 40 % of the total
C9-CI are calculated to be stabilised at 1 atm. If all of the $CH_2OO$ is assumed to be formed
stabilised (e.g. Nguyen et al., 2009) then this gives a total SCI yield of 46 %. Earlier
experimental studies have tended to determine lower total SCI yields with Hasson et al.
(2001) reporting a total SCI yield of 0.27 from measured product yields (almost entirely
nopinone) and Hatakeyama et al. (1984) reporting a total SCI yield of 0.25. Winterhalter et al.
(2000) determined a yield of 0.16 (± 0.04) for excited $CH_2OO$ from β-pinene ozonolysis,
obtained via the nopinone yield and 0.35 for the stabilised C9-CI, giving a total SCI yield of
0.51 of all the $CH_2OO$ is assumed to be stabilised. Also, experimental studies have tended to
report higher $CH_2OO$ yields (determined from measured nopinone yields) than theoretical
studies. Nguyen et al. (2009) note that this could be because nopinone can also be formed in
bimolecular reactions of SCI-4, hence experimental studies may overestimate $CH_2OO$
production. The MCMv3.3.1 incorporates a total SCI yield of 0.25 from β-pinene ozonolysis,
with a yield of stabilised C9-CI of 0.102 and a $CH_2OO$ yield of 0.148.

### 1.2.3  Limonene derived SCI

Limonene has two double bonds at which ozone can react. Theory suggests that reaction at
the endocyclic bond is more likely; Baptista et al. (2011) calculate reaction at the endo-cyclic
bond to be 84 – 94 % (dependent on the level of theory applied). Zhang et al. (2006) suggest
the reaction at the endo-cyclic double bond to be roughly 25 times faster than at the exo-
cyclic bond, i.e. leading to a branching ratio of ca. 96 % reaction at the endo bond and the





current IUPAC recommendation (IUPAC, 2013) suggests about 95 % of the primary ozone
reaction to be at the endo bond. Leungsakul et al. (2005) reported a best fit to measurements
from chamber experiments by assuming an 85 % reaction at the endo-cyclic bond and 15 % at
the exo-cyclic bond.
Ozone reaction at the endo-cyclic bond of limonene produces four different $C_{10}$ CI (Scheme
4: CI-5a, 5b, 6a, 6b). Similar to CI-1 and CI-2 from α-pinene, CI-5 is a mono-substituted CI
for which both *syn* (5a) and *anti* (5b) conformers exist, while the other, CI-6, is di-substituted,
for which two *syn*-conformers (6a and 6b) exist. Leungsakul et al. (2005) determined a total
SCI yield from limonene ozonolysis of 0.34, consisting of $CH_2OO$ (0.05), CI-7 (0.04), CI-5
(0.15) and CI-6 (0.11). Sipilä et al. (2014) determined a total SCI yield of 0.27 (± 0.12) from
indirect experiments measuring the production of $H_2SO_4$ from $SO_2$ oxidation in the presence
of the limonene-ozone system. The MCMv3.3.1 describes only reaction with ozone at the
endocyclic double bond and recommends a total SCI yield of 0.135 with stabilisation of only
the mono-substituted CI-5.
**2  Experimental**
**2.1  Experimental Approach**
The EUPHORE facility is a 200 $m^3$ simulation chamber used primarily for studying reaction
mechanisms under atmospheric boundary layer conditions. Further details of the chamber
setup and instrumentation are available elsewhere (Becker, 1996; Alam et al., 2011), and a
detailed account of the experimental procedure, summarised below, is given in Newland et al
(2015a).
Experiments comprised time-resolved measurement of the removal of $SO_2$ in the presence of
the monoterpene-ozone system, as a function of humidity. $SO_2$ and $O_3$ abundance were
measured using conventional fluorescence and UV absorption monitors, respectively; alkene
abundance was determined via FTIR spectroscopy. Experiments were performed in the dark
(*i.e.* with the chamber housing closed; $j(NO_2) \leq 10^{-6}$ $s^{-1}$), at atmospheric pressure (*ca.* 1000
mbar) and temperatures between 287 and 302 K. The chamber is fitted with large horizontal
and vertical fans to ensure rapid mixing (*ca.* 2 minutes). Chamber dilution was monitored via
the first order decay of an aliquot of $SF_6$, added prior to each experiment. Cyclohexane (*ca.*
75 ppmv) was added at the beginning of each experiment to act as an OH scavenger, such that



SO$_2$ reaction with OH was calculated to be ≤ 1 % of the total chemical SO$_2$ removal in all
experiments.
Experimental procedure, starting with the chamber filled with clean scrubbed air, comprised
addition of SF$_6$ and cyclohexane, followed by water vapour, O$_3$ (*ca.* 500 ppbv) and SO$_2$ (*ca.*
50 ppbv). A gap of five minutes was left prior to addition of the monoterpene, to allow
complete mixing. The reaction was then initiated by addition of the monoterpene (*ca.* 400
ppbv for α-pinene and β-pinene, *ca.* 200 ppbv for limonene), and reagent concentrations
followed for roughly 30 - 60 minutes; *ca.* 30 – 90 % of the monoterpene was consumed after
this time, dependent on the reaction rate with ozone. Four α-pinene + O$_3$, five β-pinene + O$_3$,
and five limonene + O$_3$ experiments, as a function of [H$_2$O], were performed in total. Each
individual run was performed at a constant humidity, with humidity varied to cover the range
of [H$_2$O] = 0.1 – 19 × 10$^{16}$ molecules cm$^{-3}$, corresponding to an RH range of 0.1 – 28 % (at
298 K). Measured increases in [SO$_2$] agreed with measured volumetric additions across the
SO$_2$ and humidity ranges used in the experiments (Newland et al., 2015a).
**2.2   Analysis**
A range of different SCI are produced from the ozonolysis of each of the three monoterpenes
(see Schemes 2 – 4), each with their own distinct chemical behaviour (*i.e.* yields, reaction
rates); it is therefore not feasible (from these experiments) to obtain data for each SCI
independently; consequently, for analytical purposes we necessarily treat the SCI population
in a simplified (lumped) manner – see Section 2.2.2.
SCI are assumed to be formed in the ozonolysis reaction with a yield ɸ (Reaction R1). They
can then react with SO$_2$, with H$_2$O, with acids formed in the ozonolysis reaction, with other
species present, or undergo unimolecular decomposition, under the experimental conditions
applied (Reactions R2 – R5). A fraction of the SCI produced reacts with SO$_2$. This fraction (*f*)
is the loss rate of the SCI to SO$_2$ ($k_2$[SO$_2$]) compared to the sum of the total loss processes for
the SCI (Equation E1) :
$$f = \frac{k_2[\mathrm{SO_2}]}{k_2[\mathrm{SO_2}] + k_3[\mathrm{H_2O}] + k_d + k_5[\mathrm{acid}] + \mathrm{L}} \qquad (E1)$$
Here, *L* accounts for the sum of any other chemical loss processes for SCI in the chamber,
with the exception of reaction with acids these loss processes are expected to be negligible, as



discussed later. After correction for dilution, and neglecting other (non-alkene) chemical sinks
for $O_3$, such as reaction with $HO_2$ (also produced directly during alkene ozonolysis (Alam et
al., 2013; Malkin et al., 2010)), which was indicated through model calculations to account
for < 0.5 % of ozone loss under all the experimental conditions, the following equation is
derived:

$$\frac{dSO_2}{dO_3} = \phi.f$$
(E2)

From Equation E2, regression of the loss of ozone ($dO_3$) against the loss of $SO_2$ ($dSO_2$) for an
experiment at a given RH determines the product $f.\phi$ at a given point in time. This quantity
will vary through the experiment as $SO_2$ is consumed, and other potential SCI co-reactants are
produced, as predicted by Equation E1. A smoothed fit was applied to the experimental data
for the cumulative consumption of $SO_2$ and $O_3$, $\Delta SO_2$ and $\Delta O_3$, (as shown in Figure 2) to
determine $dSO_2/dO_3$ (and hence $f.\phi$) at the start of each experiment, for use in Equation E2.
The start of each experiment (*i.e.* when $[SO_2] \sim 50$ ppbv) was used as this corresponds to the
greatest rate of production of the SCI, and hence largest experimental signals (*i.e.* greatest $O_3$
and $SO_2$ rate of change; greatest precision) and is the point at which the SCI + $SO_2$ reaction
has the greatest magnitude compared with any other potential loss processes for either
reactant species (see discussion below).
Other potential fates for SCIs include reaction with ozone (Kjaergaard et al., 2013; Vereecken
et al., 2014; Wei et al., 2014; Vereecken et al., 2015), with other SCI (Su et al., 2014;
Vereecken et al., 2014), carbonyl products (Taatjes et al., 2012), acids (Welz et al., 2014), or
with the parent alkene (Vereecken et al., 2014; Decker et al., 2017). Sensitivity analyses using
the most recent theoretical predictions (Vereecken et al., 2015) indicate that the reaction with
ozone may be significant under certain conditions, accounting for up to 7% of SCI loss for
*anti*-SCI (based on *anti*-$CH_3CHOO$) at the lowest RH (worst case) experiment. However,
generally SCI loss to ozone is calculated to be < 5% for *anti*-SCI and < 1% for *syn*-SCI.
Summed losses from reaction with SCI (self-reaction), carbonyls and alkenes are calculated to
account for < 1 % of the total SCI loss under the experimental conditions applied.
$CH_2OO$ and $CH_3CHOO$ have been shown to react rapidly ($k = 1 - 5 \times 10^{-10}$ cm$^3$ s$^{-1}$) with
formic and acetic acid (Welz et al., 2014). In ozonolysis experiments, Sipilä et al. (2014)
determined the relative reaction rate of acetic and formic acids with $(CH_3)_2COO$ (i.e. $k_5/k_2$) to
be roughly three. Organic acid mixing ratios in this work, as measured by FTIR, reached up to



a few hundreds of ppbv, suggesting these will likely be a significant SCI sink in our
experiments. We have therefore explicitly included reaction with organic acids in our
analysis, incorporating the uncertainty arising from the (unknown) acid reaction rate constant,
as described in Section 2.2.1.
To date, the effects of the water dimer, $(H_2O)_2$ on SCI removal have only been determined
experimentally for $CH_2OO$ (Berndt et al., 2014; Chao et al., 2015; Lewis et al., 2015;
Newland et al., 2015a). Theoretical calculations (Vereecken and Francisco, 2012) predicted
the significant effect of the water dimer compared to the monomer for $CH_2OO$, but also that
the ratio of the SCI + $(H_2O)_2$ : SCI + $H_2O$ rate constants, $k_5/k_3$, of the larger, more substituted
SCI, *anti*-$CH_3CHOO$ and $(CH_3)_2COO$, are 2 - 3 orders of magnitude smaller than for $CH_2OO$
(Vereecken and Francisco, 2012). This would make the dimer reaction negligible at
atmospherically accessible [$H_2O$] (*i.e.* $< 1 \times 10^{18}$ cm$^{-3}$) for SCI larger than $CH_2OO$. Therefore,
the effect of the water dimer reaction with $C_{10}$– and $C_9$–SCI is not considered in this analysis.
For $CH_2OO$, the reaction rates with water and the water dimer have been quantified in recent
EUPHORE experimental studies, and the values from Newland et al. (2015a) are used.
## 2.2.1 Derivation of $k$(SCI+$H_2O$)/$k$(SCI+$SO_2$) and $k_d$/$k$(SCI+$SO_2$)
As noted above, a range of different SCI are produced from the ozonolysis of the three
monoterpenes (see Schemes 2 – 4), each with their own distinct chemical behaviour, which
treated individually, introduce too many unknowns (*i.e.* yields, reaction rates) for explicit
analysis. Consequently for analytical purposes we treat the SCI population in a simplified
(lumped) manner:
Firstly, we use the simplest model possible, assuming that a single SCI is formed in each
ozonolysis reaction (Equation E3).

$$\frac{f}{[SO_2]} = \left( [SO_2] + \frac{k_3}{k_2}[H_2O] + \frac{k_d}{k_2} + \frac{k_5}{k_2}[\text{acid}] \right)^{-1}$$

(E3)
Secondly, for each monoterpene, the SCI produced are assumed to belong to one of two
populations, denoted SCI-A and SCI-B. These two populations are split according to the
observation that the decomposition rates and reaction rates with water for the smaller SCI
($CH_3CHOO$) have been predicted theoretically (Ryzhkov and Ariya, 2004; Kuwata et al.,
2010; Anglada et al., 2011) and shown experimentally (Taatjes et al., 2013; Sheps et al.,



2014; Newland et al., 2015a) to exhibit a strong dependence on the structure of the molecule.
The *syn*-CH₃CHOO conformer, which has the terminal oxygen of the carbonyl oxide moiety
in the *syn* position to the methyl group, has been shown to react very slowly with water and to
readily decompose, via the hydroperoxide mechanism; whereas the *anti*-CH₃CHOO
conformer, with the terminal oxygen of the carbonyl oxide moiety in the *anti*-position to the
methyl group, has been shown to react fast with water and is not able to decompose via the
hydroperoxide mechanism. Vereecken and Francisco (2012) have shown that all SCI studied
theoretically with an alkyl group in the *syn* position have reaction rates with $H_2O$ of $k < 4 \times$
$10^{-17}$ molecule cm³ s⁻¹ (and for SCI larger than acetone oxide, $k < 8 \times 10^{-18}$ molecule cm³ s⁻¹).
We thus define two populations, assuming SCI-A (i.e. SCI that exhibit chemical properties of
the *anti*-type SCI) to react fast with water and not to undergo unimolecular reactions, and
SCI-B (i.e. SCI that exhibit chemical properties of the *syn* type SCI) to not react with water
but to undergo unimolecular reactions. This simplification allows us to fit to the
measurements using Equations E4 and E5, as shown below. The total SCI yields are
determined by our experiments at high $SO_2$, and the relative yields of SCI-A and SCI-B are
determined from fitting to Equation E5. These relative yields are then compared to those
predicted from the literature.
In this model, $f = \gamma^A f^A + \gamma^B f^B$, where $\gamma$ is the fraction of the total SCI yield (i.e. $\gamma^A + \gamma^B = 1$). $f^A$
and $f^B$ are the fractional losses of SCI-A and SCI-B to reaction with $SO_2$. Adapting Equation
E1 to include the two SCI species gives Equation E4, where $k_5[acid]$ accounts for the SCI +
acid reaction (see discussion of reaction rate constants below).
$$f = \frac{\gamma^A k_2^A [SO_2]}{k_2^A [SO_2] + k_3 [H_2O] + k_5^A [acid]} + \frac{\gamma^B k_2^B [SO_2]}{k_2^B [SO_2] + k_d + k_5^B [acid]}$$   (E4)
Equation E4 can be rearranged to Equation E5 and fitted according to $f/[SO_2]$ derived from
the measurements.
$$\frac{f}{[SO_2]} = \frac{\gamma^A}{[SO_2] + \frac{k_3}{k_2^A}[H_2O] + \frac{k_5^A}{k_2^A}[acid]} + \frac{\gamma^B}{[SO_2] + \frac{k_d}{k_2^B} + \frac{k_5^B}{k_2^B}[acid]}$$   (E5)
Using values for $\gamma^A$ and $\gamma^B$ from the literature and varying the assumed values of the reaction
of SCI with acid ($k_5$) allows us to determine $k_3/k_2^A$ and $k_d/k_2^B$.





The assumptions made here allow analysis of a very complex system. However, a key
consequence is that the relative rate constants obtained from the analysis presented here are
not representative of the elementary reactions of any single specific SCI isomer formed, but
rather represent a quantitative ensemble description of the integrated system, under
atmospheric boundary layer conditions, which may be appropriate for atmospheric modelling.
Additionally our experimental approach cannot determine absolute rate constants (*i.e.* values
of $k_2$, $k_3$, $k_d$) in isolation, but is limited to assessing their relative values, measured under
atmospheric conditions, which may be placed on an absolute basis through use of an external
reference value (here the SCI + $SO_2$ rate constant).

## 2.2.2  SCI yield calculation

The value for the total SCI yield of each monoterpene, $\phi_{SCI\text{-}TOT}$, was determined from an
experiment performed under dry conditions (RH < 1%) in the presence of excess $SO_2$ (*ca.*
1000 ppbv), such that $SO_2$ scavenged the majority of the SCI. From Equation E2, regressing
$dSO_2$ against $dO_3$ (corrected for chamber dilution), assuming $f$ to be unity (*i.e.* all the SCI
produced reacts with $SO_2$), determines the value of $\phi_{min}$, a lower limit to the SCI yield. Figure
1 shows the experimental data, from which $\phi_{min}$ was derived.
In reality $f$ will be less than one, at experimentally accessible $SO_2$ levels, as a fraction of the
SCI may still react with trace $H_2O$ present, or undergo unimolecular reaction. The actual
yield, $\phi_{SCI}$, was determined by combining the result from the excess-$SO_2$ experiment with
those from the series of experiments performed at lower $SO_2$, as a function of $[H_2O]$, to obtain
$k_3/k_2$ and $k_d/k_2$ (see Section 2.2.1), through an iterative process to determine the single unique
value of $\phi_{SCI}$ which fits both datasets, as described in Newland et al. (2015a), but taking into
account the proposed model in this paper of there being two SCI produced. In this model, $f =$
$\gamma^A f^A + \gamma^B f^B$. Where $f^A = [SO_2] / ([SO_2] + k_3[H_2O]/k_2)$ and $f^B = [SO_2] / ([SO_2] + k_d/k_2)$ – other
possible SCI sinks are assumed to be negligible. In these excess-$SO_2$ experiments, $f^A \sim 1$ but
$f_B < 1$ since $k_d$ still represents a significant sink.
$\gamma^A$ (and hence $\gamma^B$, since $\gamma^B = 1 - \gamma^A$) is derived from fitting Equation E4 to the data from the
experiments performed at lower $SO_2$ for a given $\phi$. Using a range of $\phi$, gives a range of $\gamma$.
These different values of $\gamma$ are used with the respective values of $\phi$ in fitting to Equation E4
to determine values of $k_3/k_2$ and $k_d/k_2$.



## 3   Theoretical calculations

The rovibrational characteristics of all conformers of the CI formed from α-pinene and β-pinene, the transition states for their unimolecular reaction, and for their reaction with $H_2O$, were characterized quantum chemically, first using the M06-2X/cc-pVDZ level of theory, and subsequently refined at the M06-2X/aug-cc-pVTZ level. To obtain accurate barrier heights for reaction, it has been shown (Berndt et al., 2015; Chhantyal-Pun et al., 2017; Fang et al., 2016a, 2016b; Long et al., 2016; Nguyen et al., 2015) that post-CCSD(T) calculations are necessary. Unfortunately, performing such calculations for the SCI discussed in this paper, with up to 14 non-hydrogen atoms, is well outside our computational resources, though CCSD(T)/aug-cc-pVTZ single point energy calculations were performed for the unimolecular reactions of nopinone oxides and the most relevant subset of pinonaldehyde oxides. These data are sufficient for relative rate estimates, but it remains useful to improve the absolute barrier height predictions, using the data set by Vereecken et al. (Vereecken et al., 2017). This data set has a large number of systematic calculations on smaller CI, allowing empirical corrections to the DFT or CCSD(T) barrier heights to estimate the post-CCSD(T) barrier heights. The methodology for these corrections is described in more detail in Vereecken et al. (2017); briefly, it compares rate coefficient calculations against available harmonized experimental and very-high level theoretical kinetic rate predictions, and adjusts the barrier heights by 0.4 to 2.6 kcal mol$^{-1}$ (depending on the base methodology and the reaction type) to obtain best agreement with these benchmark results.

Using the energetic and rovibrational data thus obtained, multi-conformer transition state theory (MC-TST) calculations (Truhlar et al., 1996; Vereecken and Peeters, 2003) were performed to obtain the rate coefficient at 298K at the high pressure limit. All rate predictions incorporate tunnelling corrections using an asymmetric Eckart barrier (Eckart, 1930; Johnston and Heicklen, 1962). For the reaction of CI + $H_2O$, a pre-reactive complex is postulated at 7 kcal mol$^{-1}$ below the free reactants, while the CI + $(H_2O)_2$ reaction is taken to have a pre-reactive complex of 11 kcal mol$^{-1}$ stability. This pre-reactive complex affects tunnelling corrections; it is assumed that this pre-reactive complex is always in equilibrium with the free reactants.

In view of the high number of rotamers and the resulting computational cost, only a single limonene-derived CI isomer was studied, where the TS for the CI + $H_2O$ reaction was analyzed at the M06-2X/cc-pVDZ level of theory with only a partial conformational analysis;



a limited number of the energetically most stable TS conformers thus discovered were re-
optimized at the M06-2X/aug-cc-pVTZ level of theory. These data will only be used for
qualitative assessments. However, we apply the structure-activity relationships (SARs)
presented by Vereecken et al. (Vereecken et al., 2017) to obtain an estimate of the rate
coefficients, and assess the role of the individual SCI isomers in limonene ozonolysis.
All quantum chemical calculations were performed using Gaussian-09 (Frisch et al., 2010).

## 8    4    GEOS-Chem Model Simulation

The global chemical transport model GEOS-Chem (v9-02, www.geos-chem.org, Bey et al.,
2002) is used to explore the spatial and temporal variability of the atmospheric impacts of the
experimentally derived chemistry. The model includes HOx-NOx-VOC-$O_3$-BrOx chemistry
(Mao et al., 2010; Parrella et al., 2012) and a mass-based aerosol scheme. Biogenic
monoterpene emissions are taken from the Model of Emissions of Gases and Aerosols from
Nature (MEGAN) v2.1 inventory (Guenther et al., 2006; 2012). Transport is driven by
assimilated meteorology (GEOS-5) from NASA's Global Modelling and Assimilation Office
(GMAO). The model is run at 4°×5° resolution, with the second year (2005) used for analysis
and first year discarded as spin up.
In this study, the standard simulation was expanded to include emissions of seven
monoterpene species (α-pinene, β-pinene, limonene, myrcene, ocimene, carene, and sabinene)
from MEGAN v2.1. The ozonolysis scheme for each monoterpene, detailed in Section 6.1,
considers the formation of one or two types of SCI, and their subsequent reaction with $SO_2$,
$H_2O$, or unimolecular decomposition. Reaction rate of the monoterpenes with OH, $O_3$ and
$NO_3$ rare detailed in Table S1.

## 25   5    Results

### 26   5.1    SCI Yield

Figure 1 shows the lower limit to the SCI yield, $\phi_{min}$, for the three monoterpenes, determined
from fitting Equation E5 to the experimental data. This gives values of 0.16 (± 0.01) for α-
pinene, 0.53 (± 0.01) for β-pinene and 0.20 (± 0.01) for limonene. These $\phi_{min}$ values were
then corrected as described in Section 2.2.2 using the $k_3/k_2$ and $k_d/k_2$ values determined from





the measurements shown in Figures 3 – 5 using Equation E4. The corrected yields, $\phi_{SCI}$, are
0.19 (± 0.01) for α-pinene, 0.60 (± 0.03) for β-pinene and 0.23 (± 0.01) for limonene.
Uncertainties are ± 2σ, and represent the combined systematic (estimated measurement
uncertainty) and precision components. Literature yields for SCI production from
monoterpene ozonolysis are summarised in Table 1.
The value derived for the total SCI yield from α-pinene in this work of 0.19 agrees, within the
uncertainties, with the value of 0.15 (± 0.07) reported by Sipilä et al. (2014) and the value of
0.20 applied in the MCMv3.3.1.
The total SCI yield from β-pinene derived in this work, 0.60, agrees reasonably well with the
recent experimental work of Ahrens et al. (2014) who derived a total SCI yield of 0.50 (0.40
for the sum of CI-1 and CI-2 and 0.10 for $CH_2OO$, which is assumed to be formed almost
completely stabilised). The MCMv3.3.1 applies a total SCI yield of 0.25, of which 0.10 is a
C9-CI and 0.15 is $CH_2OO$. Earlier studies also tended to derive lower total SCI yields ranging
from 0.25 – 0.27 (Hasson et al., 2001; Hatakeyama et al., 1984).
The total SCI yield from limonene derived in this work, 0.23 (± 0.01) agrees with the recently
determined yield from Sipilä et al. (2014) of 0.27 (± 0.12). Leungsakul et al. (2005) derived a
somewhat higher yield of 0.34, while the MCMv3.3.1 applies a lower yield of 0.135.
**5.2**   $k_3(SCI+H_2O)/k_2(SCI+SO_2)$ **and** $k_d/k_2(SCI+SO_2)$ **Analysis**
Figure 2 shows the loss of $SO_2$ as ozone is consumed by reaction with the monoterpene for
each of the three systems. Box modelling results suggest that > 99 % of this $SO_2$ removal is
caused by reaction with SCI produced in the alkene-ozone reaction (rather than e.g. reaction
with OH, which is scavenged by cyclohexane). When the experiments are repeated at higher
relative humidity, the rate of loss of $SO_2$ decreases. This is as expected from Equation E1 and
suggests that there is competition between $SO_2$ and $H_2O$ for reaction with the SCI produced,
in agreement with observations of smaller SCI, which demonstrate the same competition
under atmospherically relevant conditions (Newland et al., 2015a; Newland et al., 2015b).
However, as the relative humidity is increased further, the $SO_2$ loss does not fall to (near) zero
as would be expected from Equation E1. This suggests that at high [$H_2O$] the amount of $SO_2$
loss becomes less sensitive to [$H_2O$]. This is most likely due to there being at least two
chemically distinct SCI species present. This behaviour was previously observed for



CH₃CHOO by Newland et al. (2015a) and fits with the current understanding that the
reactivity of SCI is structure dependent.
To recap Section 2.2.1, the analysis presented here considers two models to fit the
observations. The first of these (Equation E3) assumes the formation of a single SCI species,
which, in addition to reacting with $SO_2$, can react with water, undergo unimolecular reaction
or react with acid. It is clearly evident from Figures 3 – 5 that this model does not give a good
fit to the observations for any of the monoterpene systems studied. Therefore, the results from
this (single SCI) approach are not discussed explicitly hereafter. The second of the models
(Equation E5) assumes the formation of two lumped, chemically distinct, populations of SCI,
denoted SCI-A and SCI-B. SCI-A is assumed to react fast with $H_2O$ and to have minimal
decomposition. Conversely, SCI-B is assumed to have a negligible reaction with water under
the experimental conditions applied but to undergo rearrangement via a VHP. We use a least-
squares fit of Equation E5 to the data to determine the values of $k_3/k_2$ and $k_d/k_2$. This approach
fits the data well (Figures 3 - 5) for all 3 monoterpenes and represents the overall attributes of
the SCI formed - but as noted, does not represent an explicit determination of individual
conformer-dependent rate constants.

### 5.2.1 α-pinene

The α-pinene system is sensitive to water vapour at the low $H_2O$ range, with the $SO_2$ loss
falling dramatically when the RH is increased from 0.1 to 2.5 % (Figure 2). However, at
higher RH the $SO_2$ loss appears to be rather insensitive to [$H_2O$].
CI-1 can be formed in either a *syn* (1a) or *anti* (1b) configuration, whereas both CI-2
conformers formed are in a *syn* configuration (see Scheme 2). For one of the two conformers
of CI-2 (CI-2b), the hydrogen atom available for abstraction by the terminal oxygen of the
carbonyl oxide group is attached to the carbon on the four-membered ring. This has been
shown in the β-pinene system to make a large difference with respect to the ability of the
hydrogen to be abstracted and to undergo the VHP mechanism (Rickard et al., 1999; Nguyen
et al., 2009). This therefore suggests that CI-2b may exhibit characteristics of both SCI-A and
SCI-B. Ma et al. (2008) infer a probable equal yield of the two basic CI structures. This
would suggest a relative yield for SCI-A of 0.25 – 0.50 (depending on the precise nature of
CI-2b). Fitting Equation E4 to the data and allowing lambda to vary determines values of $\gamma^A$ =
0.40 and $\gamma^B$ = 0.60 (Figure 3).



In Figure 3, Equation E4 is fitted to the α-pinene measurements, assuming
$k(SCI+acid)/k(SCI+SO_2) = 0$. This derives a minimum value for $k(SCI\text{-}A+H_2O)/k(SCI\text{-}$
$A+SO_2)$, the water dependent fraction of the SCI, and a maximum value for
$k(decomposition:SCI\text{-}B)/k(SCI\text{-}B+SO_2)$, the water independent fraction of the SCI. The
kinetic parameters derived from the fitting are displayed in Table 2.
Figure 6 shows the variation of the derived $k_3/k_2$ and $k_d/k_2$ values as the ratio $k_5/k_2$,
$k(SCI+acid)/k(SCI+SO_2)$, is varied from zero to one. The derived $k_3/k_2$ increases by about 40
% from $1.4\ (\pm 0.34) \times 10^{-3}$ to $2.0\ (\pm 0.49) \times 10^{-3}$. The derived $k_d/k_2$ value decreases, again by
about 40 %, from $8.2\ (\pm 1.5) \times 10^{12}\ cm^{-3}$ to $5.1\ (\pm 0.93) \times 10^{12}\ cm^{-3}$.
The derived limits to the relative rate constants can be put on an absolute scale using the
$k(SCI+SO_2)$ values for $CH_3CHOO$ from Sheps et al. (2014) for the *syn* and *anti* conformers.
These are, *syn*: $2.9 \times 10^{-11}\ cm^3\ s^{-1}$ and *anti*: $2.2 \times 10^{-10}\ cm^3\ s^{-1}$. The *syn* rate constant is applied
to the derived $k(decomposition:SCI\text{-}B)/k(SCI\text{-}B+SO_2)$ value and the *anti* rate constant to the
$k(SCI\text{-}A+H_2O)/k(SCI\text{-}A+SO_2)$ value. It should be noted that the $k_2$ values are for quite
different SCI to those formed in this study and to our knowledge no structure specific
$k(SCI+SO_2)$ have been reported for monoterpene derived SCI, though Ahrens et al. (2014)
determine an average $k_2 \sim 4 \times 10^{-11}\ cm^3\ s^{-1}$ for SCI derived from β-pinene, i.e. a value within
an order of magnitude of those determined for the smaller SCI $CH_2OO$, $CH_3CHOO$ and
$(CH_3)_2COO$ (e.g. Welz et al., 2012; Taatjes et al., 2013; Sheps et al., 2014; Huang et al.,
2015). Using the Sheps et al. (2014) values yields $k(SCI\text{-}A+H_2O) > 3.1\ (\pm 0.75) \times 10^{-13}\ cm^3\ s^{-}$
$^{1}$ and $k(decomposition:SCI\text{-}B) < 240\ (\pm 44)\ s^{-1}$ (using the values derived for $k(SCI\text{-}$
$A+acid)/k(SCI\text{-}A+SO_2) = 0$). This $k_3$ value is an order of magnitude larger than the rate
constants determined for the smaller *anti*-$CH_3CHOO$ in the direct studies of Sheps et al.
(2014) $(2.4 \times 10^{-14}\ cm^3\ s^{-1})$ and Taatjes et al. (2013) $(1.0 \times 10^{-14}\ cm^3\ s^{-1})$. The decomposition
value derived for SCI-B is of the same order of magnitude as that for *syn*-$CH_3CHOO$ ($348 \pm$
$332\ s^{-1}$) and $(CH_3)_2COO$ ($819 \pm 190\ s^{-1}$) from Newland et al., (2015a) (using updated direct
measurement values of $k_2$ from Sheps et al. (2014) and Huang et al. (2015) for *syn*-
$CH_3CHOO$ and $(CH_3)_2COO$ respectively) and within the range from the recent paper by
Smith et al. (2016) which derives a decomposition rate for $(CH_3)_2COO$ of $269\ (\pm 82)\ s^{-1}$ at
283 K increasing to $916\ (\pm 56)\ s^{-1}$ at 323 K.
Sipilä et al. (2014) applied a single-SCI analysis approach to the formation of $H_2SO_4$ from
$SO_2$ oxidation in the presence of the α-pinene ozonolysis system. They determined that for α-





pinene, $k_d \gg k(SCI+H_2O)[H_2O]$ for $[H_2O] < 2.9 \times 10^{17}$ cm$^{-3}$, i.e. that the fate of SCI formed
in the system is rather insensitive to $[H_2O]$. Across the $[SO_2]$ and RH ranges used in their
study, the results obtained here would indicate $H_2O$ to always be the dominant sink for SCI-
A, i.e. the fact that Sipilä et al. (2014) see similar $H_2SO_4$ production across the RH range in
their study is consistent with these results.

### 5.2.2  β-pinene

Two recent studies (Nguyen et al., 2009; Ahrens et al., 2014) have suggested yields of the two
$C_9$-CI (CI-3 and CI-4, see Scheme 3) obtained from β-pinene ozonolysis to be roughly equal.
In these studies Ahrens et al. (2014) assume a $CH_2OO$ yield of 0.10 while Nguyen et al.
(2009) determine theoretically the yield of $CH_2OO$ to be 0.05. Another theoretical study
(Zhang and Zhang, 2005) predicted a $CH_2OO$ yield of 0.08. In experimental studies,
Winterhalter et al. (2000) determined the $CH_2OO$ yield to be 0.16 (± 0.04) from measuring
the nopinone yield and assuming it to be entirely a primary ozonolysis product (i.e. the co-
product of $CH_2OO$ formation) and Ma and Marston (2008) determine a summed contribution
of 84 % (± 0.03) for the two $C_9$-CI (i.e. a 16 % $CH_2OO$ yield). The theoretical studies are
somewhat lower than the experimental but Nguyen et al. (2009) note that CI-4 is likely to
form additional nopinone in bimolecular reactions. The $CH_2OO$ is assumed to all be formed
stabilised (e.g. Nguyen et al. 2009).
SCI-3 is expected to undergo unimolecular reactions at least an order of magnitude faster than
SCI-4 (Nguyen et al., 2009; Ahrens et al., 2014). The reaction of SCI-3 with water is expected
to be slow based on the calculations presented in Table 4, with a pseudo first order reaction
rate of 1.0 s$^{-1}$ at 75 % RH, 298 K, whereas the water reaction with SCI-4 is expected to be
considerably faster with a pseudo first order reaction rate of 240 s$^{-1}$ at 75 % RH, 298 K. This
reaction will thus likely be the dominant fate of SCI-4 at typical atmospheric RH. This is in
agreement with the observations of Ma and Marston (2008), that show a clear dependence of
nopinone formation on RH (presumed to be formed from SCI + $H_2O$). Fitting Equation E4 to
the data determines values of $\gamma^A = 0.41$ and $\gamma^B = 0.59$ (Figure 4).
Using these values, and assuming $k(SCI+acid)/k(SCI+SO_2) = 0$, yields a $k(SCI-$
$A+H_2O)/k(SCI-A+SO_2)$ value of $> 1.0 (\pm 0.27) \times 10^{-4}$ and a $k(decomposition:SCI-B)/k(SCI-$
$B+SO_2)$ value of $< 6.0 (\pm 1.3) \times 10^{12}$ cm$^{-3}$ (Table 2).





As shown in Figure 6, increasing $k_5/k_2$, $k(SCI+acid)/k(SCI+SO_2)$, from zero to one, decreases
the derived $k_d/k_2$ from 6.0 ($\pm$ 1.3) $\times$ 10$^{12}$ cm$^{-3}$ to 1.8 ($\pm$ 0.39) $\times$ 10$^{12}$ cm$^{-3}$. The derived $k_3/k_2$
increases by a factor of four from 1.0 ($\pm$ 0.27) $\times$ 10$^{-4}$ to 3.7 ($\pm$ 1.0) $\times$ 10$^{-4}$.
These values can be put on an absolute scale (using the values derived above for $k_5/k_2 = 0$).
For SCI-A, $k(SCI+SO_2)$ is taken as the experimentally determined value of 4 $\times$ 10$^{-11}$ cm$^3$ s$^{-1}$
from Ahrens et al. (2014). For SCI-B, the $syn$-CH$_3$CHOO $k(SCI+SO_2)$ value determined by
Sheps et al. (2014) is used. This gives values of $k(SCI\text{-}A+H_2O) > 4 \times 10^{-15}$ ($\pm$ 1) cm$^3$ s$^{-1}$ and
$k(\text{decomposition:SCI-B}) < 170$ ($\pm$ 38) s$^{-1}$.

### 5.2.3 Limonene

For the limonene measurements presented in Figure 2, $(dSO_2/dO_3)/dt$ appears to be non-
linear, with a jump in $dSO_2/dO_3$ between 120 and 150 ppbv of ozone consumed. This is most
evident in the two lowest RH runs (0.2 and 2.0 %). Limonene is the fastest reacting of the
systems presented here, with the alkene reaction having consumed 100 ppbv of ozone within
the first five minutes. The limonene sample required about five minutes of heating before the
entire sample was volatized and injected into the chamber. This therefore may account for the
apparent non-linear nature of $dSO_2/dO_3$ in Figure 2.
The SO$_2$ loss in the limonene-ozone system is less affected by increasing H$_2$O than for either
$\alpha$ or $\beta$-pinene (Figure 5), with the values of $f/[SO_2]$ (y-axis) varying by roughly a factor of two
over the RH range applied compared to more than a factor of three variation for the other two
systems. Hence it might be expected that there is little formation of H$_2$O dependent SCI or
that it has a rather slow reaction rate with water.
Fitting Equation E4 to the data determines values of $\gamma^A = 0.22$ and $\gamma^B = 0.78$ (Figure 5). This
is broadly in line with the ratio recommended in the MCMv3.3.1 of 0.27:0.73, and with that
proposed in Leungsakul et al. (2005) who use a CI-A:CI-B ratio of 0.35:0.65, but also include
some stabilisation of CH$_2$OO and C$_9$-CI from ozone reaction at the exo-cyclic bond. This
yields a $k(SCI\text{-}A+H_2O)/k(SCI\text{-}A+SO_2)$ value of $< 3.5$ ($\pm$ 0.20) $\times$ 10$^{-5}$ and a
$k(\text{decomposition:SCI-B})/k(SCI\text{-}B+SO_2)$ value of $> 4.5$ ($\pm$ 0.10) $\times$ 10$^{12}$ cm$^{-3}$.
Figure 6 shows that the derived $k_d/k_2$ increases by about 7 % as $k(SCI+acid)/k(SCI+SO_2)$
ranges from 0.0 to 0.8. The derived $k_3/k_2$ becomes negative at $k(SCI+acid)/k(SCI+SO_2) > 0.8$,
putting an upper limit on this ratio, i.e. $k_5/k_2 < 0.8$, for the limonene system.





Putting these values on an absolute scale (using the values derived for $k_5/k_2 = 0$), using the
$CH_3CHOO$ *syn* and *anti* $k(SCI+SO_2)$ determined by Sheps et al. (2014), yields values of < 7.7
($\pm 0.60$) $\times 10^{-15}$ cm$^3$ s$^{-1}$ and > 130 ($\pm 3$) s$^{-1}$ for $k_3$ and $k_d$ respectively. These values are similar
to those derived for the SCI-A and SCI-B formed from β-pinene. The $k_3$ value is a factor of
three smaller than that determined by Sheps et al. (2014) for $k_3$(*anti*-$CH_3CHOO+H_2O$), 2.4 $\times$
$10^{-14}$ cm$^3$ s$^{-1}$.
Sipilä et al. (2014) applied a single-SCI analysis approach to the formation of $H_2SO_4$ from
$SO_2$ oxidation by the limonene ozonolysis system and determined that, similarly to α-pinene,
$k$(decomp.) >> $k$(SCI+ $H_2O$)[$H_2O$] for [$H_2O$] < 2.9 $\times 10^{17}$ cm$^{-3}$, i.e. that the system is rather
insensitive to [$H_2O$]. Our data are consistent with the limonene system being less sensitive to
[$H_2O$] than the SCI populations derived from the other two monoterpenes reported here.

### 12   5.2.4 Experimental Summary

The removal of $SO_2$ in the presence of ozonolysis reactions of α-pinene, β-pinene and
limonene has been studied as a function of water vapour concentration, and analysed
following the approximation that the SCI population can be represented through a two species
model, with contrasting unimolecular decomposition rates and reactivity to water. The results
presented in this work suggest that all three monoterpenes studied produce a range of SCI that
have differing reactivities towards water and decomposition. This is in agreement with current
theoretical understanding but is the first experimental demonstration for large SCI derived
from monoterpene ozonolysis. The complex reactivity of the systems investigated is further
highlighted by the fact that the experimental data are not fitted well by the assumption of the
formation of a single SCI species. While the behaviour of large SCI derived from
monoterpenes are likely to be significantly more complicated than is accounted for by simply
considering the differing kinetics of *syn* and *anti* SCI conformers, this approach provides a
reasonable description of the experimental behaviour observed, and the results presented here
are broadly in line with experimental results from the smaller SCI and from theoretical
results. The reaction rates of SCI-A (i.e. SCI that exhibit chemical properties of the *anti*-type
SCI) derived from the three different monoterpenes with water range from < 0.8 to > 31 $\times 10^{-}$
$^{14}$ cm$^3$ s$^{-1}$, broadly in line with the derived rates of Sheps et al. (2014) for *anti*-$CH_3CHOO$ of
2.4 $\times 10^{-14}$ cm$^3$ s$^{-1}$. The decomposition rates of SCI-B (i.e. SCI that exhibit chemical
properties of the *syn*-type SCI) are on the order of 100 - 250 s$^{-1}$. This is in line with those
derived for *syn*-$CH_3CHOO$ from *cis* and *trans*-but-2-ene ozonolysis and $(CH_3)_2COO$ by



Newland et al. (2015a) of 348 ($\pm$ 332) s$^{-1}$ and 819 ($\pm$ 190) s$^{-1}$ respectively (assuming $k$(*syn*-
CH$_3$CHOO+SO$_2$) = 2.9 $\times$ 10$^{-11}$ cm$^3$ s$^{-1}$ (Sheps et al., 2014) and $k$((CH$_3$)$_2$COO+SO$_2$) = 2.9 $\times$
10$^{-10}$ cm$^3$ s$^{-1}$ (Huang et al., 2015)) and recent results from Smith et al. (2016) of 269 – 916 s$^{-1}$
(strongly dependent on temperature) for (CH$_3$)$_2$COO decomposition. In this work we only
derive relative rates, but the similarity of the $k_3$ and $k_d$ values derived when the $k_2$ values for
*syn* and *anti*-CH$_3$CHOO from Sheps et al. (2014) are applied is consistent with the recent
work of Ahrens et al. (2014), suggesting that large SCI, derived from monoterpenes,
demonstrate a similar reactivity towards SO$_2$ as smaller SCI. One uncertainty in the derivation
of the kinetics presented herein is the reactions of the SCI produced with organic acids. These
acids were present in the experiments (owing to formation in the monoterpene ozonolysis
reactions themselves) at levels which may have been a competitive sink for the SCI.
The ability of the simplified SCI-A / SCI-B approach to fit the experimental data and the
good agreement with theory and experimental work for smaller SCI suggests that the kinetic
parameters derived herein, using a lumped two-SCI system, may be useful for modelling and
provide the best available basis for modelling the effects of SCI on atmospheric SO$_2$
oxidation in the presence of water vapour. To this end, in Section 6 we present the results of a
global modelling study using the kinetic parameters derived herein.
### 5.3  Theoretical results and comparison to experiments
The theoretically predicted rate coefficients for unimolecular reactions of the monoterpene
SCI are listed in Table 3, while those for the reaction with H$_2$O are listed in Table 4. These
data can be compared against the experimental data obtained in this work.
### 5.3.1  $\alpha$-pinene
The theory-based rate coefficients show one pinonaldehyde oxide, CI-1b, with a rate of
reaction with water that is significantly faster than the remaining $\alpha$-pinene-derived CI.
Comparing this rate to the experimental data suggests that CI-1b corresponds to SCI-A, with
matching rate coefficients within an order of magnitude, i.e. within the expected uncertainty.
We thus deduce that SCI-A is CI-1b. The remaining pinonaldehyde oxides, CI-1a, CI-2a and
CI-2b, react predominantly through unimolecular reactions, where theory-based rate
coefficients range from 60 to 600 s$^{-1}$, all within a factor of 4 of the experimentally derived
population-averaged rate of 240 $\pm$ 44 s$^{-1}$, i.e. matching within the uncertainty margins. The





unimolecular rate coefficients of this set of CI are sufficiently close that it is not feasible to
separate these in the experimental data, so we can only conclude that SCI-B in the α-pinene
ozonolysis experiments may consist of a mixture of C-1a, CI-2a and CI-2b.
### 5.3.2 β-pinene
The theoretical analysis for nopinone oxides shows one isomer, SCI-4, that has a fast rate of
reaction with water, but a slow unimolecular isomerisation, while the other isomer, SCI-3,
shows a fast unimolecular decomposition. These can thus be unequivocally equated to the
experimentally obtained SCI-A and SCI-B, respectively, inasmuch as the yield of $CH_2OO$ is
minor. The predicted rate coefficients are within the expected uncertainty intervals of the
theoretical data, a factor of 5 for the unimolecular rates, and an order of magnitude for the
reaction with $H_2O$.
The experimental rate measurements are defined relative to the reaction rate with $SO_2$; the
value adopted for the $k(SCI+SO_2)$ reaction therefore influences the derived rate coefficient
values. Ahrens et al. (2014) directly measured the $SO_2$ rate coefficient of the longest-lived
SCI (SCI-4) to be $\sim 4 \times 10^{-11}$ $cm^3$ $s^{-1}$, but for SCI-3 we assume a similar rate coefficient as
$syn$-$CH_3CHOO$ + $SO_2$ determined by Sheps et al. (2014) of $2.9 \times 10^{-11}$ $cm^3$ $s^{-1}$. Nopinone
oxides are bicyclic compounds, with a bulky dimethyl-substituted 4-membered ring adjacent
to the carbonyl oxide moiety. To examine the potential impact of steric hindrance on the SCI
+ $SO_2$ reaction, we characterized all sulfur-substituted secondary ozonides (S-SOZ) formed in
this reaction (Kuwata et al., 2015; Vereecken et al., 2012). We find that the tri-cyclic S-SOZ
shows very little interaction between the sulfur-bearing ring and the β-pinene substituents,
and little change in ring strain. The energies of the S-SOZ adducts relative to the SCI + $SO_2$
reactants thus remains very similar to that of $CH_2OO$, $CH_3CHOO$ or $(CH_3)_2COO$, confirming
the quality of our selection of reference rate coefficients.
### 5.3.3 Limonene
Of the six non-$CH_2OO$ CI formed in limonene ozonolysis, CI-5b was predicted to have a fast
reaction rate with $H_2O$; its oxide substitution patterns is similar to pinonaldehyde oxide CI-1b.
The SAR-predicted rate coefficient of CI-5b + $H_2O$ is within a factor of 2 of the
experimentally derived $k_3$ value for SCI-A, such that we can equate SCI-A to CI-5b with
confidence. The SCI-B set of Criegee intermediates then contains the summed population of
the remaining five CI, all of which react slowly with $H_2O$. The SAR-predicted unimolecular





decay rate coefficients range from 15 to 700 s$^{-1}$, all within a factor of 9 of the experimentally
obtained $k_d$ = 130 s$^{-1}$; it should be noted that for limonene-derived CI, no explicit theoretical
calculations are available, and the SAR-predictions carry a somewhat larger uncertainty.
We have performed an exhaustive characterisation of the conformers of CI-5b. The most
stable conformers show an internal complex formation between the oxide moiety and the
carbonyl group, similar to those characterized for the bimolecular reaction of CI with
carbonyl compounds (Jalan et al., 2013; Wei et al., 2015). The theoretical study by Jiang et al.
(2013) on limonene ozonolysis appears to have omitted internal rotation and cannot be
compared directly. It seems likely that the limonene-derived CI can thus easily undergo
internal SOZ formation, which is thought (Vereecken and Francisco, 2012) to be entropically
unfavourable, but to have a low barrier to reaction. For α-pinene, a similar internal complex
formation and SOZ ring closure is not as favourable due to the geometric limitations enforced
by the 4-membered ring.
A large number of transition state conformers for CI-5b + H$_2$O were characterized, though no
exhaustive search was completed. The energetically most favourable structures show
interaction between the carbonyl group, and the H$_2$O co-reactant as it adds onto the carbonyl
oxide moiety. Similar stabilising interactions between the carbonyl moiety and the
carbonyl oxide moiety were reported recently in cyclohexene-derived CI
(Berndt et al., 2017). This interaction thus lowers the barrier to reaction though it is currently
unclear whether it enhances the reaction rate compared to e.g. the α-pinene-derived CI-1b, as
these hydrogen-bonded structures are entropically not very favourable. The intra-molecular
interactions with heterosubstituents could be investigated in future work.

## 24 6 Global modelling study

### 25 6.1 SCI Chemistry

A global atmospheric modelling study was performed using the GEOS-Chem chemical
transport model (as described in Section 4) to examine the global monoterpene derived SCI
budget and the contribution of these SCI to gas-phase SO$_2$ oxidation. The existing chemistry
scheme in the model is supplemented with monoterpene SCI chemistry based on the
experimental results described in Section 5 and in Table 5. It should be noted here that this
modelling study focuses on the chemical impacts of monoterpene SCI formed from





ozonolysis reactions only. No chemistry for other SCIs derived from isoprene and/or other
(smaller) alkenes are incorporated in the adapted model chemical scheme used.
The monoterpene emissions in GEOS-Chem are taken from MEGAN v2.1 (Guenther et al.,
2012). The scheme emits seven monoterpenes: α-pinene, β-pinene, limonene, myrcene,
ocimene, 3-carene, and sabinene. The monoterpenes are oxidised within the model by OH,
$NO_3$ and $O_3$ at rates shown in Table S1. Reaction with $O_3$ leads to the production of
monoterpene specific SCI. Reactions with OH and $NO_3$ does not lead to the formation of any
products, with the reactions only acting as a sink for the monoterpene and the respective
oxidant. The SCI yields from the ozonolysis of α-pinene, β-pinene, and limonene are derived
from the experimental work presented here. SCI from each monoterpene are split in to SCI-A
and SCI-B as defined in previous sections. For the other four monoterpenes emitted, the SCI
yields, and kinetics are derived based on similarity of structure to one of the species studied
here or previously in the literature. The main SCI produced in the ozonolysis of myrcene and
ocimene are expected to be acetone oxide ($(CH_3)_2COO$) or 4-vinyl-5-hexenal oxide
($CH_2CHC(CH_2)CH_2CH_2CHOO$), since ozone has been suggested to react predominantly at
the isolated internal double bond (~97 % for myrcene, ~90% for ocimene (Baker et al.,
2004)). The SCI yield is taken to be 0.30, similar to that of $(CH_3)_2COO$ from 2,3-dimethyl-
but-2-ene ozonolysis (Newland et al., 2015a). However, this may be an underestimate since it
has been predicted that stabilisation of small CI increases with an increasing size of carbonyl
co-product, as this co-product can take more of the nascent energy of the primary ozonide on
decomposition due to a greater number of degrees of freedom available (Nguyen et al., 2009,
Newland et al., 2015b). Sabinene is a bicyclic monoterpene with an external double bond and
hence is treated like β-pinene. This assumption is backed up by recent theoretical work (Wang
and Wang, 2017), who predict similar behaviour of sabinene derived SCI to the predicted
behaviour of β-pinene SCI by Nguyen et al. (2009a). They predict a SCI yield between 24 % -
64 %. 3-carene is a bicyclic monoterpene with an internal double bond and is treated like α-
pinene.
**6.2   Modelling Results**
Figure 7 shows the annually averaged total SCI burden from monoterpene ozonolysis in the
surface layer in the GEOS-Chem simulation. A number of interesting features are apparent
from this figure and the associated information given in Table 6:





(i)      The highest annually averaged monoterpene SCI concentrations are found
2            above tropical forests.

(ii)      Peak anually averaged monoterpene SCI concentrations are $\sim 1.2 \times 10^4$ $cm^{-3}$.
(iii)      > 97 % of the total monoterpene SCI burden is SCI-B.
Annual global monoterpene emissions are dominated by the tropics (Figure S1), accounting
for > 90 % during the northern hemisphere winter months (November – April) and 70 % even
during the peak emissions from the northern boreal region during June and July (Sindelarova
et al., 2014). Despite annually averaged surface ozone mixing ratios being roughly a factor of
2 higher in the northern mid-high latitudes, monoterpene SCI production is still dominated by
the tropics. Annually averaged surface monoterpene SCI concentrations across the northern
boreal regions are $< 2 \times 10^3$ $cm^{-3}$; during the summer months (JJA) this value rises to $2 - 5 \times$
$10^3$ $cm^{-3}$.
More than 97 % of the total monoterpene derived SCI are SCI-B (Table 6). This is because
typical water vapour concentrations in the tropics are $> 5.0 \times 10^{17}$ $cm^{-3}$. This gives SCI-A
removal rates (i.e. $k_3[H_2O]$) of $2 \times 10^3 - 1.5 \times 10^5$ $s^{-1}$, whereas removal rates of SCI-B to
unimolecular reactions have been determined here to be $1 - 3$ orders of magnitude slower, on
the order of 100 - 250 $s^{-1}$. Since the loss of SCI-B is independent of temperature in the model,
the highest SCI-B concentrations would be expected to be located in the regions of highest
SCI-B production. Recent experimental studies (Smith et al., 2016) have demonstrated a
strong temperature dependence for the unimolecular decomposition rate of $(CH_3)_2COO$
between 283 and 323 K (269 – 916 $s^{-1}$). Therefore, it may be that in reality there would be
some geographical variation in the rate of unimolecular loss.
The monoterpene SCI-A + $H_2O$ reactions are expected to lead to high yields of both large
(e.g. Ma et al., 2008; Ma and Marston, 2008) and small (measured in high yield in the
experiments presented here) organic acids.
Figure 8 shows the seasonal removal of $SO_2$ by reaction with monoterpene derived SCI, as a
percentage of total gas-phase $SO_2$ oxidation in the surface layer. Monoterpene SCI are most
important (relative to OH) for $SO_2$ oxidation over tropical forests, where they account for up
to 50 % of the local gas-phase $SO_2$ removal during DJF and MAM in some regions. The
reasons for this are two-fold: firstly, the highest modelled monoterpene SCI concentrations
are found in these regions (Figure 7); but additionally, OH concentrations in the model are
low over these areas (Figure S2). Historically there has been discrepancies between modelled





and observed OH concentrations over tropical forests, with models appearing to under-predict
[OH] by up to a factor of ten (e.g. Lelieveld et al., 2008). It was proposed that this was due to
missing sources of OH recycling during isoprene oxidation. During recent years there have
been advances in our understanding of isoprene chemistry. GEOS-Chem v-09, used here,
includes an isoprene OH recycling scheme largely based on Paulot et al. (2009a, 2009b), with
updates from Peeters et al. (2009), Peeters and Müller (2010), and Crounse et al. (2011;
2012), and evaluated in Mao et al. (2013). However, more recent experimental and theoretical
work is not yet included.
Annually, monoterpene SCI oxidation accounts for 1.1 % of the gas-phase $SO_2$ oxidation in
the terrestrial tropics. This accounts for the removal of 2.5 Gg of $SO_2$. Across the northern
boreal forests, monoterpene SCI contribute 0.5 % to gas-phase $SO_2$ removal annually,
removing 0.6 Gg of $SO_2$. Globally, throughout the whole atmosphere, monoterpene SCI
account for only 0.4 % of gas-phase $SO_2$ removal, removing 6.8 Gg of $SO_2$ annually.
It is noted that MEGAN does not contain oceanic monoterpene emissions, which may
increase the global importance of SCI for gas-phase $SO_2$ removal. Luo and Yu (2010)
determined annual global oceanic α-pinene emissions to be 29.5 TgC using a top-down
approach, with only 0.013 (Luo and Yu, 2010) – 0.26 (Hackenberg et al., 2017) TgC
estimated using a range of bottom-up approaches; clearly there are large uncertainties in
oceanic monoterpene emissions. At the upper end of this range they could potentially provide
a similar contribution to SCI production and subsequent $SO_2$ oxidation as monoterpenes
emitted from the terrestrial biosphere. SCI production more generally could be further
amplified by sources such as marine-derived alkyl iodine photolysis.
Blitz et al. (2017) recently calculated a revised $SO_2$ + OH reaction rate ($k_1$ (1 bar $N_2$) (298 K)
= 5.8 × $10^{-13}$ $cm^3$ $s^{-1}$), based on experimental work and a master equation analysis, which is ~
40 % lower than the rate given in the most recent JPL data evaluation (Burkholder et al.,
2015) (($k_1$ (1 bar $N_2$) (298 K) = 9.5 × $10^{-13}$ $cm^3$ $s^{-1}$), which is used in the GEOS-Chem model
simulation.  Figure S3 shows the increased influence of monoterpene derived SCI on gas-
phase $SO_2$ oxidation if the alternative $SO_2$ + OH rate is used. This increased the impact of
monoterpene SCI to up to 60 % of gas-phase $SO_2$ removal in regions of the tropical forests
during DJF and MAM, with the contribution of monoterpene SCI to global gas-phase $SO_2$
oxidation increasing to 0.6 %.




While certain monoterpenes appear to be more important than others with regard to the production of SCI which will oxidise SO$_2$, these results are sensitive to the kinetics used and the assumptions made for the monoterpenes not studied experimentally here. Hence we do not attempt to draw any conclusions about the relative importance of each monoterpene from the modelling. Clearly the most important monoterpenes will be those with high yields of SCI-B, particularly if those SCI-B have a structure that hinders unimolecular decomposition (such as certain β-pinene derived SCI).

## 7    Discussion and Atmospheric Implications

Monoterpene ozonolysis produces a diverse range of SCIs, with contrasting fates in the atmosphere, dominated by unimolecular reaction or reaction with water vapour, but which may still affect atmospheric SO$_2$ processing.  Monoterpene-derived SCI have the potential to make a significant contribution to gas-phase SO$_2$ oxidation in specific local (i.e. forested) environments, of up to 50 % at certain times of year - amplifying sulfate aerosol formation, reducing the atmospheric lifetime and hence geographic distribution of SO$_2$, however the results presented here show that their impact upon annual SO$_2$ oxidation globally is modest. The results presented here demonstrate that it is important that monoterpene ozonolysis reactions are considered to produce at least two different SCI species if their chemistry is to be adequately represented in global models. This is because even a 'moderate' reaction rate with water would be a dominant sink of an SCI with the averaged properties of SCI-A and SCI-B.

SCI concentrations are expected to vary greatly depending on the local environment and time of year, *e.g.* monoterpene abundance may be considerably higher (and with a different reactive mix of alkenes giving a range of structurally diverse SCI) in a forested environment, compared to a rural background. Furthermore, biogenic isoprene and monoterpene emissions are strongly temperature dependent, hence are predicted to change significantly in the future as a response to a changing climate and other environmental conditions (Peñuelas and Staudt, 2010).

This study shows that the ozonolysis of monoterpenes may contribute to significant SCI concentrations in forested areas. Another group of compounds produced by forests that may also have the potential to be a significant source of SCI are sesquiterpenes (C$_{15}$H$_{24}$). Although



generally present at low mixing ratios, this is due to their short atmospheric lifetimes caused
by their rapid reaction rates with ozone. The flux through the alkene-ozone reaction for fast
reacting monoterpenes and sesquiterpenes is often higher than for monoterpenes with high
mixing ratios but low removal rates, e.g. α-pinene and β-pinene. Ozonolysis of sesquiterpenes
has been shown to have very high SCI yields (Beck et al., 2011; Yao et al., 2014) and these
SCI have been shown to react with SCI scavengers (e.g. $SO_2$, $H_2O$ etc.) in a similar way to
smaller SCI (Yao et al., 2014). It has been predicted that SCI from sesquiterpenes may have a
high degree of secondary ozonide formation (Chuong et al., 2004) but experimental work has
shown very different results for structurally different sesquiterpenes studied (Beck et al.,
2011; Yao et al., 2014) hence this is highly uncertain, as is the fate of the SOZ once formed.
Therefore, these have the potential to be another significant source of SCI.
**8    Conclusions**
We report results from an integrated experimental (simulation chamber), theoretical (quantum
chemical) and modelling (global chemistry-transport simulation) study of the impacts of
monoterpene ozonolysis reactions on stabilised Criegee intermediate (SCI) formation and $SO_2$
oxidation.  The ozonolysis of the monoterpenes α-pinene, β-pinene and limonene have been
shown to produce a structurally diverse range of chemically distinct SCIs, with some showing
limited sensitivity to / reaction with water vapour under near-atmospheric humidity levels. A
multi-component system is required to explain the experimentally observed $SO_2$ removal
kinetics. A two-body model system based on the assumption of a fraction of the SCI produced
being reactive towards water (SCI-A; potentially contributing to the significant formation of a
range of organic acids in the atmosphere), and a fraction being relatively unreactive towards
water (SCI-B), analogous to the structural dependencies observed for the simpler $CH_3CHOO$
SCI system, has been shown to describe the observed kinetic data reasonably well for all the
monoterpene systems investigated, and may form a computationally affordable and
conceptually accessible basis for the description of this chemistry within atmospheric models.
The atmospheric fate of SCI-B produced from the monoterpenes studied here will be
controlled by their removal by unimolecular decomposition. In this work, we have
experimentally determined the monoterpene SCI-B decomposition rate to be between 100 and
250 $s^{-1}$. This has significant implications for the role of monoterpene derived SCI as oxidants
in the atmosphere. The fate of SCI-A will be reaction with water or the water dimer, likely
leading to the production of a range of organic acids.



A theory-based analysis of the kinetics of the SCI formed from α-pinene, β-pinene ozonolysis
has also been performed, which complements the experimental work. The identification of the
likely SCI-A and SCI-B populations and the derived kinetics agree with experimental
observations within the respective uncertainties.
A modelling study using the GEOS-Chem global 3-D chemical transport model supplemented
with the chemical kinetics elucidated in this work suggests that the global monoterpene
derived SCI burden will be dominated (> 97%) by SCI-B. The highest annually averaged SCI
concentrations are found in the tropics, with seasonally averaged monoterpene SCI
concentrations up to $1.2 \times 10^4$ cm$^{-3}$ owing to large monoterpene emissions. Across the boreal
forest, average SCI concentrations reach between $3 - 5 \times 10^3$ cm$^{-3}$ during the northern
hemisphere summer. Oxidation of $SO_2$ by monoterpene SCI is shown to also be most
important in the tropics. While oxidation by SCI contributes < 1% to gas-phase $SO_2$ oxidation
globally, over tropical forests this can rise to up to 50 % at certain times of the year.
Monoterpene SCI driven $SO_2$ oxidation will increase the production of sulfate aerosol –
affecting atmospheric radiation transfer, and hence climate; and reduce the atmospheric
lifetime and hence transport of $SO_2$. These effects will be substantial in areas where
monoterpene emissions are significant, in particular over the Amazon, Central Africa and SE
Asian rainforests.

**Data Availability**

Experimental data will be made available in the Eurochamp database (www.eurochamp.org)
from the H2020 EUROCHAMP2020 project, GA nº730997

**Acknowledgements**

The assistance of the EUPHORE staff is gratefully acknowledged., Salim Alam, Marie
Camredon and Stephanie La are thanked for helpful discussions. This work was funded by
EU FP7 EUROCHAMP 2 Transnational Access activity (E2-2012-05-28-0077) and the UK
NERC Projects (NE/K005448/1, Reactions of Stabilised Criegee Intermediates in the
Atmosphere: Implications for Tropospheric Composition & Climate) and (NE/M013448/1,
Mechanisms for Atmospheric chemistry: GeneratioN, Interpretation and FidelitY -





MAGNIFY). Fundación CEAM is partly supported by Generalitat Valenciana, and the
project DESESTRES (Prometeo Program - Generalitat Valenciana). EUPHORE
instrumentation is partly funded by the Spanish Ministry of Science and Innovation, through
INNPLANTA project: PCT-440000-2010-003. LV is indebted to the Max Planck Graduate
Center with the Johannes Gutenberg-Universität Mainz (MPGC).





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



1    Table 1. Monoterpene SCI yields derived in this work and reported in the literature.

| $\phi_{SCI}$ | Reference | Notes | Methodology |
|---|---|---|---|
| **α-pinene** | | | |
| 0.19 (± 0.01) | This work | | $SO_2$ loss |
| 0.15 (± 0.07) | Sipilä et al. (2014) | | Formation of $H_2SO_4$ |
| 0.22 | Taipale et al. (2014) (personal comm. Berndt) | | |
| 0.125 (± 0.04) | Hatakeyama et al. (1984) | | Formation of $H_2SO_4$ |
| 0.20 | MCMv3.3.1 [a] | | |
| **β-pinene** | | | |
| 0.60 (± 0.03) | This work | | $SO_2$ loss |
| 0.46 | Ahrens et al. (2014) | $\varphi_{C9-SCI}$: 0.36 $\varphi_{CH2OO}$: 0.10 | FTIR detection |
| 0.25 | MCMv3.3.1 [a] | $\varphi_{C9-SCI}$: 0.102 $\varphi_{CH2OO}$: 0.148 | |
| 0.42 | Nguyen et al. (2009) | $\varphi_{C9-SCI}$: 0.37 $\varphi_{CH2OO}$: 0.05 | Theoretical |
| 0.51 | Winterhalter et al. (2000) | $\varphi_{C9-SCI}$: 0.35 $\varphi_{CH2OO}$: 0.16 | Change in nopinone yields $f([H_2O])$ |
| 0.44 | Kotzias et al. (1990) | | Formation of $H_2SO_4$ |
| 0.25 | Hatakeyama et al. (1984) | | Formation of $H_2SO_4$ |
| 0.30 | Zhang and Zhang (2005) | $\varphi_{C9-SCI}$: 0.22 $\varphi_{CH2OO}$: 0.08 | |
| > 0.27 | Ma and Marston (2008) | $\varphi_{C9-SCI}$: 0.27 $\varphi_{CH2OO}$: 0.16[a] $\varphi_{CH2OO}$: 0.06[b] | Change in nopinone yields $f([H_2O])$ |
| 0.27 | Hasson et al. (2001) | | Change in nopinone yields $f([H_2O])$ |
| **Limonene** | | | |
| 0.23 (± 0.01) | This work | | $SO_2$ loss |
| 0.27 (± 0.12) | Sipilä et al. (2014) | | Formation of $H_2SO_4$ |
| 0.34 | Leungsakul et al. (2005) | $\varphi_{C10-SCI}$: 0.26 $\varphi_{CI-x}$: 0.04 $\varphi_{CH2OO}$: 0.05 | Measurement of stable particle and gas-phase products |
| 0.135 | MCMv3.3.1 [a] | | |



Uncertainty ranges (± 2σ, parentheses) indicate combined precision and systematic measurement error
components for this work, and are given as stated for literature studies. All referenced experimental studies
produced SCI from MT + $O_3$ and were conducted between 700 and 760 Torr. [a] http://mcm.leeds.ac.uk/MCM/
(Jenkin et al., 2015).
[a] assuming 100 % stabilisation
[b] assuming 40 % stabilisation





Table 2. Monoterpene derived SCI relative and absolute[a] rate constants derived in this work.

| SCI | $10^5 \, k_3/k_2$ | $10^{15} \, k_3$ ($cm^3 \, s^{-1}$) | $10^{-12} \, k_d/k_2$ ($cm^{-3}$) | $k_d$ ($s^{-1}$) |
|---|---|---|---|---|
| **α-pinene** | | | | |
| SCI-A | > 140 (± 34) | > 310 (± 75)[a] | | |
| SCI-B | | | < 8.2 (± 1.5) | < 240 (± 44)[c] |
| **β-pinene** | | | | |
| SCI-A | > 10 (± 2.7) | > 4 (± 1)[b] | | |
| SCI-B | | | < 6.0 (± 1.3) | < 170 (± 38)[c] |
| **Limonene** | | | | |
| SCI-A | < 3.5 (± 0.2) | < 7.7 (± 0.6)[a] | | |
| SCI-B | | | > 4.5 (± 0.1) | > 130 (± 3)[c] |

Uncertainty ranges (± 2σ, parentheses) indicate combined precision and systematic measurement error
components. [a] Scaled to an absolute value using $k_2$(anti-CH$_3$CHOO) = 2.2 × 10$^{-10}$ cm$^3$ s$^{-1}$ (Sheps et al., 2014); [b]
Scaled to an absolute value using $k_2$(anti-CH$_3$CHOO) = 4 × 10$^{-11}$ cm$^3$ s$^{-1}$ (Ahrens et al., 2014); [c] Scaled using
$k_2$(syn-CH$_3$CHOO) = 2.9 × 10$^{-11}$ cm$^3$ s$^{-1}$ (Sheps et al., 2014).



Table 3. Unimolecular reactions for the CI derived from α-pinene, β-pinene, and *d*-limonene,
as derived by Vereecken et al. (2017). Barrier heights (kcal mol$^{-1}$) listed estimate post-
CCSD(T) energies.

| Carbonyl oxide | Reaction | $E_b$ | $k$(298K) / s$^{-1}$ |
|---|---|---|---|
| ***α-pinene*** | | | |
| CI-1a | 1,4-H-migration | 15.8 | 600 |
| | SOZ-formation | 15.6 | $5 \times 10^{-2}$ |
| | 1,3-ring closure | 21.6 | $1 \times 10^{-3}$ |
| CI-1b | 1,3-ring closure | 14.8 | 60 |
| | 1,3-H-migration | 29.0 | $1 \times 10^{-6}$ |
| CI-2a | 1,4-H-migration | 16.3 | 250 |
| | 1,3-ring closure | 20.8 | $6 \times 10^{-3}$ |
| CI-2b | 1,4-H-migration | 17.0 | 60 |
| | SOZ-formation | 13.5 | 8 |
| | Ring closure | 19.9 | $3 \times 10^{-2}$ |
| ***β-pinene*** | | | |
| CI-3 | 1,4-H-migration | 15.7 | 375 |
| | 1,3-ring closure | 21.1 | $2 \times 10^{-3}$ |
| CI-4 | 1,3-ring closure | 17.2 | 2.0 |
| | Ring opening | 23.6 | (Slow, Nguyen et al. 2009a) |
| | 1,4-H-migration | 24.9 | (Slow, Nguyen et al. 2009a) |
| CH$_2$OO | 1,3-ring closure | 19.0 | 0.3 |
| | 1,3-H-migration | 30.7 | $1 \times 10^{-7}$ |
| ***Limonene*[a]** | | | |
| CI-5a | 1,4-H-migration | SAR | 200 [a] |
| CI-5b | 1,3-ring closure | SAR | 75 [a] |
| CI-6a | 1,4-H-migration | SAR | 430 [a] |
| CI-6b | 1,4-H-migration | SAR | 700 [a] |
| CI-7a | 1,4-H-migration | SAR | 15 |
| CI-7b | 1,4-H-migration | SAR | 600 |

[a] Formation of secondary ozonides (SOZ) is not included, and could be the dominant unimolecular loss.



Table 4. Rate coefficients ($cm^3$ molecule$^{-1}$ s$^{-1}$) for the reaction of CI with $H_2O$ and $(H_2O)_2$ as
predicted by Vereecken et al. (2017). Values are based on explicit CCSD(T)/aug-cc-
pVTZ//M06-2X/aug-cc-pVTZ calculations and multi-conformer TST, including empirical
corrections to reference experimental data, except for limonene-derived CI where the values
are predicted using a structure-activity relationship. The rate coefficients for $CH_2OO$,
$CH_3CHOO$, and $(CH_3)_2COO$ are within a factor of 4 of evaluated literature data (Vereecken et
al., 2017).

| Carbonyl oxide | $k$(298K) $H_2O$ | $k$(298K) $(H_2O)_2$ |
|---|---|---|
| $CH_2OO$ | $8.7 \times 10^{-16}$ | $1.4 \times 10^{-12}$ |
| *syn*-$CH_3CHOO$ | $6.7 \times 10^{-19}$ | $2.1 \times 10^{-15}$ |
| *anti*-$CH_3CHOO$ | $2.3 \times 10^{-14}$ | $2.7 \times 10^{-11}$ |
| $(CH_3)_2COO$ | $7.5 \times 10^{-18}$ | $1.8 \times 10^{-14}$ |
| *α-pinene* | | |
| CI-1a | $1.3 \times 10^{-18}$ | $2.9 \times 10^{-15}$ |
| CI-1b | $1.5 \times 10^{-14}$ | $1.7 \times 10^{-11}$ |
| CI-2a | $1.0 \times 10^{-18}$ | $2.5 \times 10^{-15}$ |
| CI-2b | $2.4 \times 10^{-19}$ | $7.0 \times 10^{-16}$ |
| *β-pinene* | | |
| CI-3 | $1.7 \times 10^{-18}$ | $4.3 \times 10^{-15}$ |
| CI-4 | $4.2 \times 10^{-16}$ | $6.4 \times 10^{-13}$ |
| *Limonene* | | |
| CI-5a | $1.5 \times 10^{-18}$ | $4.3 \times 10^{-15}$ |
| CI-5b | $1.5 \times 10^{-14}$ | $1.7 \times 10^{-11}$ |
| CI-6a | $9.1 \times 10^{-18}$ | $2.1 \times 10^{-14}$ |
| CI-6b | $1.5 \times 10^{-17}$ | $3.2 \times 10^{-14}$ |
| CI-7a | $9.7 \times 10^{-18}$ | $1.9 \times 10^{-14}$ |
| CI-7b | $4.3 \times 10^{-18}$ | $1.1 \times 10^{-14}$ |





Table 5. Kinetic parameters used in the global modelling study.

| SCI | $\phi_{SCI}$ | $10^{15} k_3$ (cm$^3$ s$^{-1}$) | $10^{11} k_2$ [a] (cm$^3$ s$^{-1}$) | $k_d$ (s$^{-1}$) |
|---|---|---|---|---|
| **_α-pinene_** | | | | |
| SCI-A | 0.08 | 310 | 22 | - |
| SCI-B | 0.11 | - | 2.9 | 240 |
| **_β-pinene_** | | | | |
| SCI-A | 0.25 | 4 | 4 | - |
| SCI-B | 0.35 | - | 2.9 | 170 |
| **_Limonene_** | | | | |
| SCI-A | 0.05 | 7.7 | 22 | - |
| SCI-B | 0.18 | - | 2.9 | 130 |
| **_Myrcene_** | | | | |
| SCI-B | 0.30 | - | 13[b] | 819[c] |
| **_Ocimene_** | | | | |
| SCI-B | 0.30 | - | 13[b] | 819[c] |
| **_Sabinene_** [d] | | | | |
| SCI-A | 0.25 | 4 | 4 | - |
| SCI-B | 0.35 | - | 2.9 | 170 |
| **_3-carene_** [e] | | | | |
| SCI-A | 0.08 | 310 | 22 | - |
| SCI-B | 0.11 | - | 2.9 | 240 |

[a] $k_2$(SCI-A+SO$_2$) from (SO$_2$+_anti_-CH$_3$CHOO) - Sheps et al. (2014); $k_2$(SCI-B+SO$_2$) from (SO$_2$+_syn_-CH$_3$CHOO) - Sheps et al. (2014) unless otherwise stated

[b] $k_2$(SCI-B+SO$_2$) from (SO$_2$+_anti_-(CH$_3$)$_2$COO) – Huang et al. (2015)

[c] $k_d$(SCI-B) from Newland et al. (2015) (scaled to $k_2$(SCI-B+SO$_2$) from Huang et al. (2015)

[d] Kinetics based on _β_-pinene

[e] Kinetics based on _α_-pinene



1  Table 6. Monoterpene contribution to [SCI] and SO$_2$ oxidation in the surface layer of the
2  model simulation.

| Monoterpene | Annual emissions[a] (Tg C) | % contribution to [SCI-A] | % contribution to [SCI-B] | % contribution to SO$_2$ oxidation |
|---|---|---|---|---|
| α-pinene | 35.4 | 0.5 | 16 | 6.9 |
| β-pinene | 16.9 | 74 | 46 | 65 |
| limonene | 9.2 | 3.5 | 14 | 7.2 |
| myrcene | 3.1 | 0.0 | 1.2 | 4.5 |
| trans-β-ocimene | 14.1 | 0.0 | 5.4 | 11 |
| sabinene | 7.9 | 22 | 14 | 4.5 |
| 3-carene | 6.4 | 0.0 | 2.7 | 1.6 |

3  [a] From MEGAN v2.1 (Guenther et al., 2012)



**Primary
Ozonide**

3    Scheme 1. Simplified generic mechanism for the reaction of Criegee Intermediates (CIs)

4    formed from alkene ozonolysis.

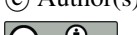



Scheme 2. Mechanism of formation of the two Criegee Intermediates (CIs) from α-pinene
ozonolysis.



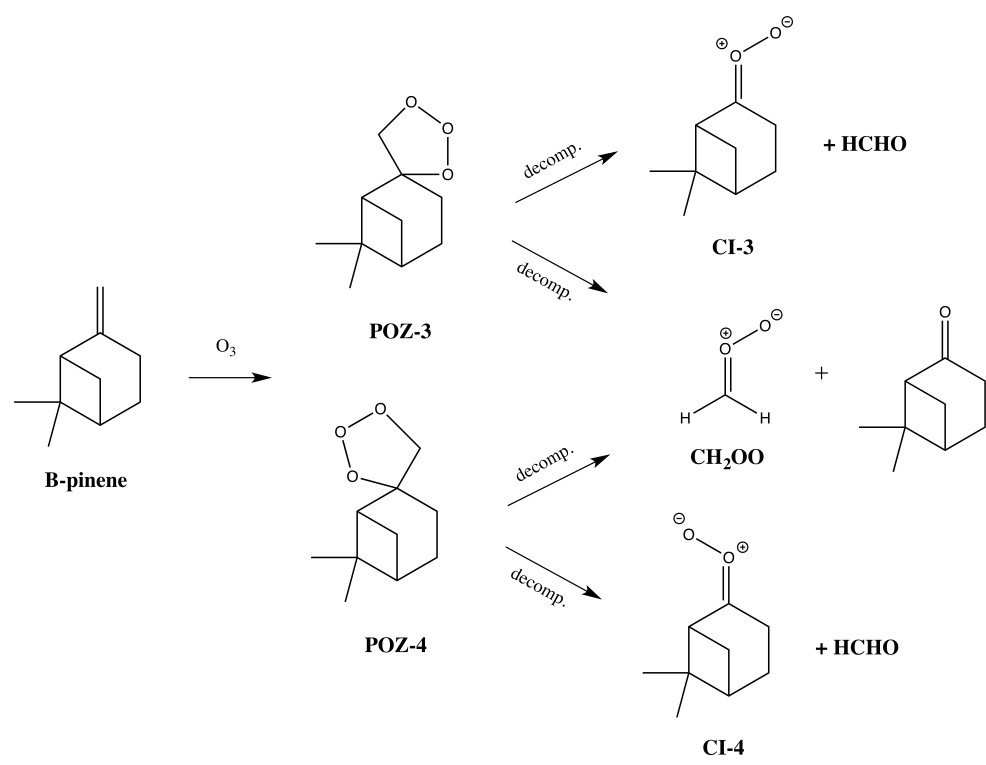

3    Scheme 3. Mechanism of formation of the three Criegee Intermediates (CIs) from β-pinene

4    ozonolysis.





3    Scheme 4. Mechanism of formation of the four Criegee Intermediates (CIs) from limonene

4    ozonolysis.





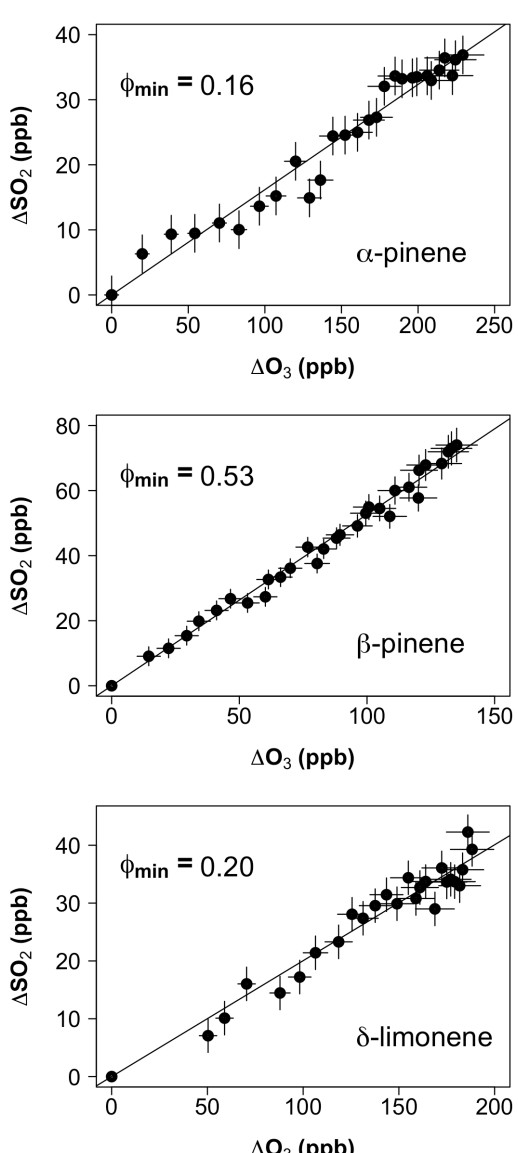

2    Figure 1. $\Delta SO_2$ vs. $\Delta O_3$ during excess $SO_2$ experiments ($[H_2O] < 5 \times 10^{15}$ cm$^{-3}$). The gradient

3    determines the minimum SCI yield ($\phi_{min}$).



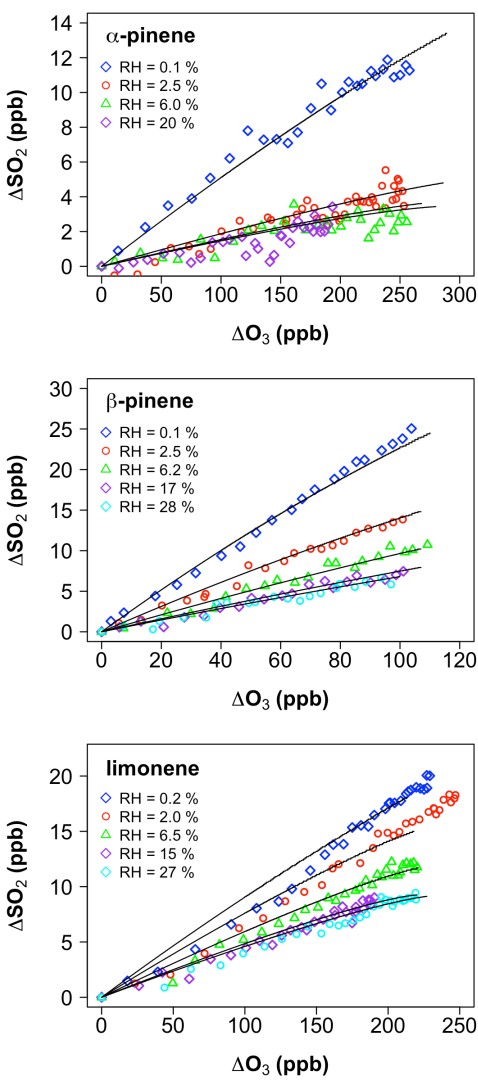

Figure 2. Cumulative consumption of SO$_2$ as a function of cumulative consumption of O$_3$,
ΔSO$_2$ versus ΔO$_3$, for the ozonolysis of α-pinene, β-pinene and limonene in the presence of
SO$_2$ at a range of water vapour concentrations, from $1 \times 10^{15}$ cm$^{-3}$ to $1.9 \times 10^{17}$ cm$^{-3}$. Symbols
are experimental data, corrected for chamber dilution. Lines are smoothed fits to the
experimental data.




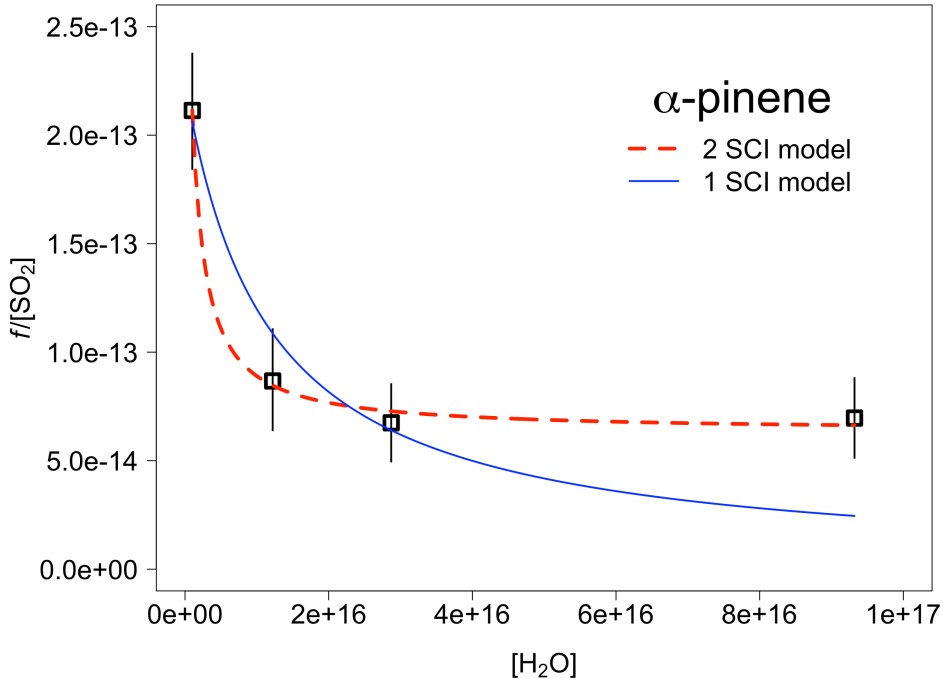

2 Figure 3. Application of a 2 SCI model fit (Equation E4) and a single SCI model fit (Equation

3 E1) to the measured values (open squares) of $f$/[SO$_2$] for α-pinene. From the fit we derive

4 relative rate constants for reaction of the α-pinene derived SCI, SCI-A and SCI-B with H$_2$O

5 ($k_3/k_2$) and decomposition (($k_d$+$L$)/$k_2$) assuming that $\gamma^A$ = 0.40 and $\gamma^B$ = 0.60.





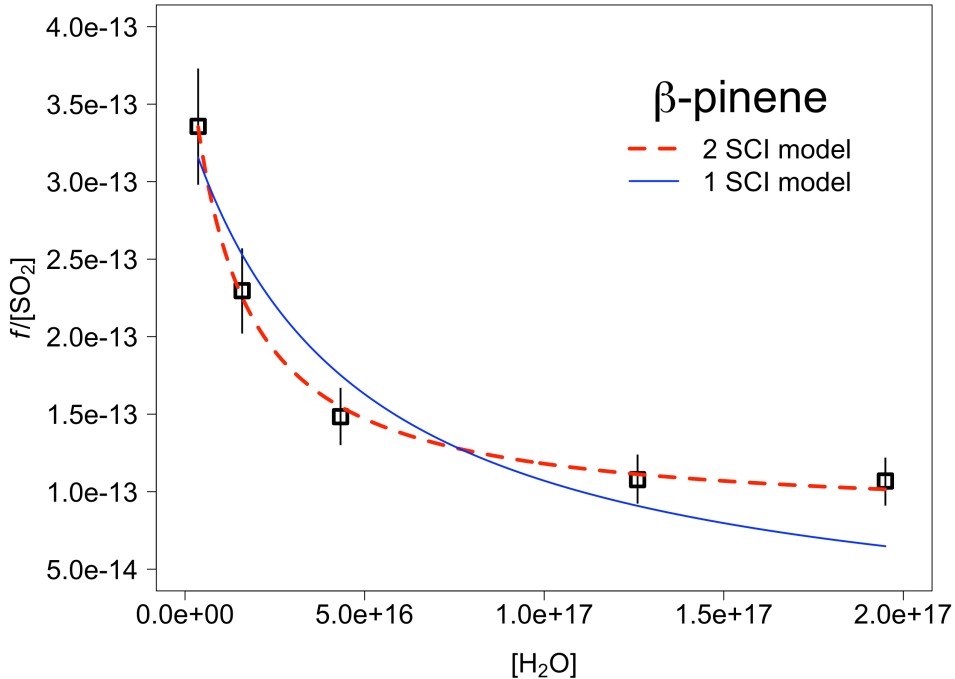

Figure 4. Application of a 2 SCI model fit (Equation E4) and a single SCI model fit (Equation
E1) to the measured values (open squares) of $f$/[SO$_2$] for β-pinene. From the fit we derive
relative rate constants for reaction of the β-pinene derived SCI, SCI-A and SCI-B with H$_2$O
($k_3/k_2$) and decomposition (($k_d$+$L$)/$k_2$) assuming that $\gamma^A = 0.41$ and $\gamma^B = 0.59$.




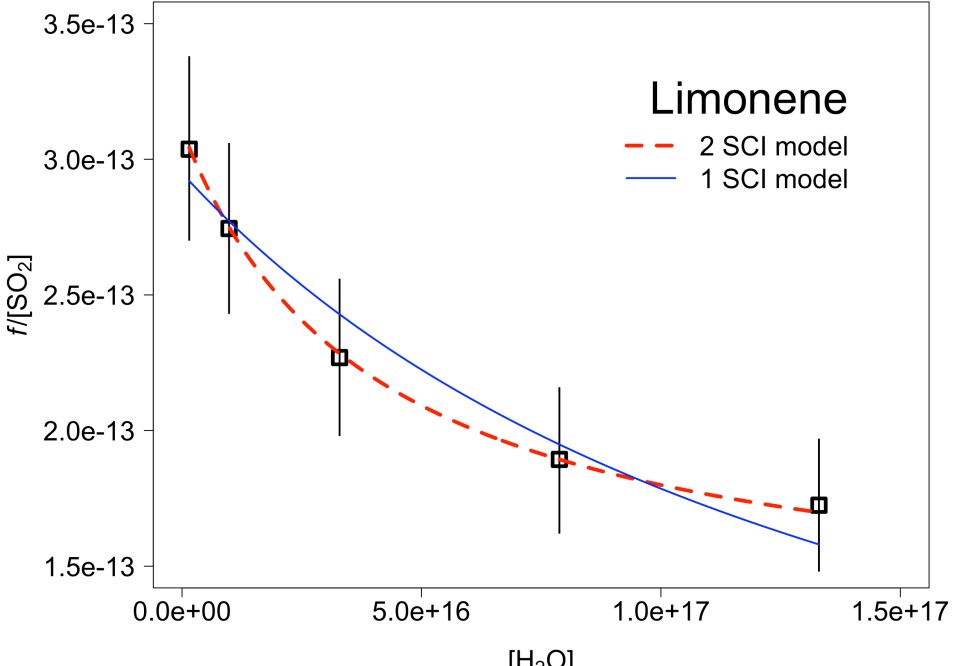

Figure 5. Application of a 2 SCI model fit (Equation E4) and a single SCI model fit (Equation
E1) to the measured values (open squares) of $f$/[$SO_2$] for limonene. From the fit we derive
relative rate constants for reaction of the limonene derived SCI, SCI-A and SCI-B with $H_2O$
($k_3/k_2$) and decomposition (($k_d+L$)/$k_2$) assuming that $\gamma^A = 0.22$ and $\gamma^B = 0.78$.





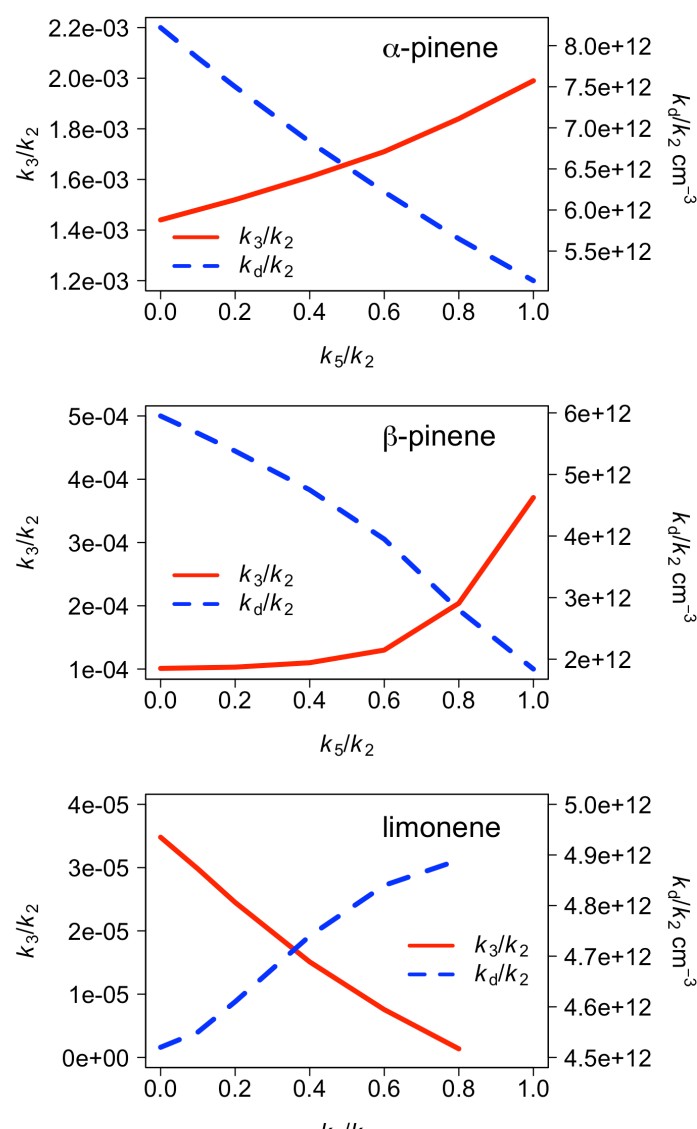

Figure 6. Variation of $k_3/k_2$ ($k$(SCI-A+$H_2O$))/($k$(SCI-A+$SO_2$)) and $k_d$ ($k$(SCI-B
unimol.))/($k$(SCI-B+$SO_2$)) as a function of the ratio $k_5/k_2$ ($k$(SCI+acid)/$k$(SCI+$SO_2$)), derived
from least squares fit of Equation E4 to measurements shown in Figures 3 -5 for α-
pinene, β-pinene and limonene respectively.



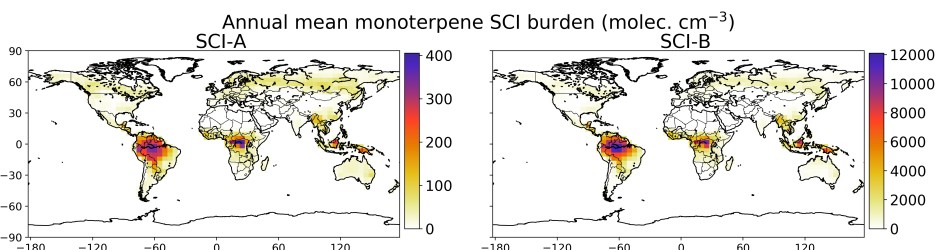

Figure 7. Annual mean monoterpene SCI-A and SCI-B concentrations (cm$^{-3}$) in the surface
layer of the GEOS-Chem simulation.

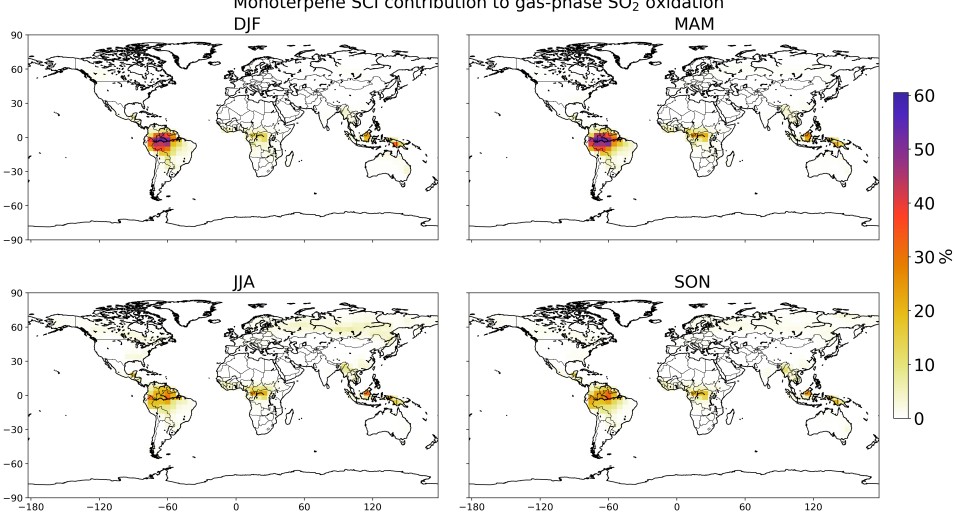

2 Figure 8. Seasonal SO$_2$ oxidation by monoterpene SCI as percentage of total gas-phase SO$_2$
3 oxidation in the surface layer.
4
