# Peer review of "The atmospheric impacts of monoterpene ozonolysis on"

_Atmospheric Chemistry and Physics, 2017_

## Referee Comment (RC1) · Anonymous Referee #1 · 28 Dec 2017

In this paper, the authors performed a chamber experiment to understand the effect of humidity toward the SO2 reaction of the ozonolysis product of the alpha-, and beta-pinene; and limonene. The 298 K experimental result was analyzed assuming the reaction of two different types of stabilized Criegee intermediate (SCI) forming from the ozonolysis. SCIA which favorably reacts with H2O and SCIB which favorably unimolecularly decomposes. This two SCI model provided a good fit to the chamber results. Using this fits, they can obtain relative ratios between SCI bimolecular reaction rates with atmospheric trace gases. Results of quantum chemistry calculations are presented

to provide support for the rates obtained from the experimental fits. These results are placed into the global simulation model to show that the SO2 reaction only accounts for <1% of the global decay path for SCI obtained from ozonolysis of monoterpene. These experimental results confirm the theoretical prediction presented by one of the authors (PCCP, 19, 31599, 2017). All in all, I think these results are important and should be published. However, I have the following points I think the authors should clarify.

First, the authors should clarify the difference between the real atmospheric environment and their chamber. The real atmospheric environment has a range of temperature, relative humidity, and pressure; they should say which temperature, humidity, pressure range can be attained in their chamber. Second, on page 22, lines 13-16, the authors mention the issues concerning the non-linear results for the limonene results in Figure 2. Can the authors cool the limonene before entering the chamber to decrease the ozonolysis rate? Or can they try an experiment at lower temperatures, to obtain cleaner data for low humidity? Third, cyclohexane is used as an OH scavenger. Is the SCI reaction with cyclohexane slow that it will not interfere with their analysis?

Small points that can be fixed are as follows: In page 13 lines 11-13, the authors mention that water dimer reaction will be negligible at atmospherically accessible [H2O]. However, it has already been shown experimentally that for anti-CH3CHOO water vapor reaction, water dimer reaction will dominate the room temperature reaction at a relative humidity (RH) above 30%. (PCCP, 18, 28189-, 2016) On the other hand, the present chamber experiments were done at RH 0.1 to 28%. Therefore, the authors should change this part to "For the analysis of the present chamber results the water dimer reaction can be ignored." In Page 21 lines 23-24, they mention the effective rates for the SCI water vapor reaction at RH 75%, 298 K, and discuss results, but their experimental chamber results are up to RH 28%, so I am not sure it is relevant to mention the results for such high RH.

[Figure]

2017.

---

## Referee Comment (RC2) · Anonymous Referee #2 · 3 Jan 2018

Manuscript no. acp-2017-1095

The authors describe experimental results from ozonolysis reactions of three monoterpens carried out in the Valencia chamber. Chosen reactant concentrations were at least partly orders of magnitude higher than atmospheric levels. Progress of the reaction for different water vapour concentrations was followed by monitoring the disappearance of $SO_2$ and $O_3$. $SO_2$ served as a Criegee intermediate (CI) scavenger. This manuscript represents a continuation of the work of this group in the field of research on the CI reactivity for close to atmospheric conditions.

The authors determined the overall fractions of collisional stabilized CIs for the different monoterpenes. Stabilized CIs from each reaction system were grouped in two different CI-proxies with either *syn*- or *anti*-behaviour regarding their chemical reactivity. Observed overall relative rate coefficients were set on an absolute scale of rate coefficient for the *syn*- or *anti*-proxies each using some simplifications as well absolute rate coefficients from literature. Based on these data, runs for global modelling of monoterpene-derived CIs importance as possible atmospheric oxidant were conducted. As a result of that a maximum steady-state CI concentration of about $10^4$ molecules $cm^{-3}$ in the tropics was found.

The manuscript is well written and contains a lot of important information. I recommend publication in ACP because it is a significant contribution for a better understanding of the role of biogenics in the atmospheric oxidation system. Some explanations and clarifications could further improve the quality of this manuscript. Here my comments:

1) All the experimental findings are based on $SO_2$ and $O_3$ measurements. Please provide more information how it was done and what´s the accuracy, detection limit etc.
2) p.15 line 13: $SO_2$ was taken in excess for CI titration. It is stated "$SO_2$ scavenged the majority of the SCI." Why the authors did not chose perfect experimental conditions for these titration experiments allowing a direct determination of the sCI fraction without any further processing of the primary data?
3) p.17/18 and table 1: Finally stated sCI yields have a quite low range of uncertainty. Does the uncertainty really reflect the overall precision of this experimental approach?
4) p.20 line 10: The authors used Sheps´s *syn*- *anti*-$CH_3CHOO$ rate coefficients to set their relative values on an absolute scale, Sheps et al., PCCP (2014). Especially the k-value of *anti*-$CH_3CHOO+SO_2$ is significantly different compared with that by Taatjes et al., Science (2013). Is there a special reason using the Sheps et al. values? What are the consequences if the Taatjes et al. data are used instead of those by Sheps et al.?

---

## Referee Comment (RC3) · Anonymous Referee #3 · 7 Jan 2018

General This is a very interesting and comprehensive study on Criegee chemistry related to monoterpene ozonlysis.

Schloary presentation: The text on monoterpene ozonolysis in the early part of the manuscript is a very nice and thorough summary but when read the reader is asking himself: 'And what is the outcome of the present paper for this ?' - this is then treated in the results section. Maybe some on the contents of the introductory text can be shortened and be used when the results are actually presented. That would also compact the paper to some extend.

Shortening certain sections and avoiding doubling of text appears advisable as the manuscipt reads kind of lengthy at times. There is the danger to loose the reader.

The theoretical chemistry section of the paper might be problematic, but I am not an expert in this.

Overall, the MS represents a big effort to better understand terpene-derived SCI atmospheric chemistry and its implications which in principle very well merits publication in ACP. However, the presentation and organisation of the manuscript should be improved. Overall, it seems revision is more in the direction of mayor rather than minor.

Details

Page 4, line 4: The population of CIs is formed...pls check sentence.

p12, l6: This equation looks strangely formatted. Pls check.

p13-l5 - 15: I feel this is partly repeating material already given in the introductory overview. That should be avoided. Please check and discuss the state-of-the art regarding the water reaction, the roles of the water dimer and the difference of syn- and anti- conformers once in the manuscipt and then work with internal referencing.

p16: If it has been shown, that post-CCSD(T) calculations are needed but these cannot be performed for technical reason, what is then the use of this ? It is difficult to judge how valida such calculations could be. Certain journals do not accept theoretical chemistry calculation not being performed with the best available techniques. The authors should deal with this. Maybe it is better to outsource this part and do the bigger calculations separately.

p18,l 29: Pls check sentence

p23, Is that section 5.2.4. really needed ? I think it should be skipped in order to streamline the whole paper.

p24, section 5.3: See general comment on this. Is it necessary to give all the structural

data in the SI ?

p 26, Why is section 6 separate from the 'results' section - these are also results, so it might be sensible to make this a sub-point of the results section 5 rather than a new section 6

p29, l 14: Oceanic MT emissions are expected to be small compared to the continental ones.

p30, sections 7 & 8: Maybe these sections can be combined.

---

## Short Comment (SC1) · 19 Jan 2018

Please could the authors clarify why a value of 819 ($\pm$190) s$^{-1}$ is being used for (CH3)2COO unimolecular reaction rate coefficient? The recent IUPAC task group on atmospheric chemical kinetic data evaluation's preferred value is 397 s$^{-1}$ at 298 K. Is the authors' global modelling study affected if more accurate rate coefficients are used?

---

## Author Comment (AC1) · 7 Mar 2018

**Response to referees of submission of:**

**The atmospheric impacts of monoterpene ozonolysis on global stabilised Criegee intermediate budgets and SO2 oxidation: experiment, theory and modelling by Newland et al., 2017, submitted to ACPD**

**General Response**

We thank the referees for giving their time to make insightful comments, helping to clarify and further improve our manuscript. All 3 referee's recognise the importance of the results presented, and recommend publication in ACP.

A couple of significant changes to note are:

(i)     The removal of part of Section 5.2.4, 'Experimental Summary' and all of Section 7, 'Discussion and Atmospheric Implications', as requested by reviewer #3. No information has been lost from the manuscript, these sections were, as pointed out by reviewer #3, somewhat repeating previous sections, a little of Section 7 has been merged into the Conclusions.

(ii)    The use of the IUPAC recommended rate coefficient for the decomposition rate of $(CH_3)_2COO$, as recommended by R. Chhantyal-Pun in a Comment. This has tended to increase the burden of SCI-B in our global modelling study, and increase the removal of $SO_2$ by SCI by ~ 20%.

See the replies to the specific reviewer for further details of these changes.

Responses to specific points raised by each reviewer are given separately beneath that point. Referees comments are bold and italic, the author's comments are inset in plain type.

**Anonymous Referee #1**

*First, the authors should clarify the difference between the real atmospheric environment and their chamber. The real atmospheric environment has a range of temperature, relative humidity, and pressure; they should say which temperature, humidity, pressure range can be attained in their chamber.*

> We agree that we could be clearer that the results are applicable to the atmospheric boundary layer (i.e. surface pressure). This is where the chemistry is important as alkene concentrations are low outside the boundary layer due to their short atmospheric lifetimes. We have clarified this in the abstract by amending the fifth sentence (P2, L2-6) to read:
>
> *"We have investigated the removal of $SO_2$ by SCI formed from the ozonolysis of three atmospherically important monoterpenes ($\alpha$-pinene, $\beta$-pinene and limonene) in the*

*presence of varying amounts of water vapour in large-scale simulation chamber experiments, representative of boundary layer conditions."*

EUPHORE is an outdoor environmental chamber and as such we have no control over the temperature. It is stated that temperature varied between 287 – 302 K across the experiments (p10, l28).

We clearly state the relative humidity of each experiment (as is required - the whole point of the experiments is to determine the effect of RH), as well as giving the overall range in the experimental section (p11, l12).

***Second, on page 22, lines 13-16, the authors mention the issues concerning the non-linear results for the limonene results in Figure 2. Can the authors cool the limonene before entering the chamber to decrease the ozonolysis rate? Or can they try an experiment at lower temperatures, to obtain cleaner data for low humidity?***

Unfortunately we are unable to perform further experiments at EUPHORE as they were a part of the REACT-SCI campaign in 2013. The possible issue with the non-linearity of the limonene loss is not the reaction rate per se, but more the low volatility of the limonene precursor, meaning that a period of a few minutes is required to add the compound to the chamber. Cooling the sample would only exacerbate this effect.

As stated above, EUPHORE is an outdoor environmental chamber and hence we have no control over the chamber temperature.

***Third, cyclohexane is used as an OH scavenger. Is the SCI reaction with cyclohexane slow that it will not interfere with their analysis?***

Reaction rates of SCI with alkanes are thought to be very slow. Recent theoretical work (Xu et al., 2017) has calculated reaction rates of $CH_2OO$ with cyclohexane to be 5.7 x $10^{-22}$ $cm^3$ $s^{-1}$. At 75 ppmv cyclohexane, as employed in our experiments, this would lead to loss rates for $CH_2OO$ to the alkane on the order of 1 x $10^{-16}$ $s^{-1}$, eight orders of magnitude lower than typical loss rates to decomposition or reaction with water.

***Small points that can be fixed are as follows:***

***In page 13 lines 11-13, the authors mention that water dimer reaction will be negligible at atmospherically accessible [H2O]. However, it has already been shown experimentally that for anti-CH3CHOO water vapor reaction, water dimer reaction will dominate the room temperature reaction at a relative humidity (RH) above 30%. (PCCP, 18, 28189-, 2016) On the other hand, the present chamber experiments were done at RH 0.1 to 28%. Therefore, the authors should change this part to "For the analysis of the present chamber results the water dimer reaction can be ignored."***

The referee is right to point out the work of Lin et al. (2016) on *anti*-CH3CHOO + $(H_2O)_2$. We have altered this paragraph in response to this comment. We have moved the first

part of the paragraph to the introduction in response to a comment from referee #2. We now include the following paragraph in the introduction:

*"To date, the effects of the water dimer, $(H_2O)_2$ on SCI removal have only been determined experimentally for $CH_2OO$ (Berndt et al., 2014; Chao et al., 2015; Lewis et al., 2015; Newland et al., 2015a; Sheps et al., 2017; Liu et al., 2017) and anti-$CH_3CHOO$ (Lin et al., 2016). Theoretical calculations (Vereecken et al., 2017) have predicted the ratio of the SCI + $(H_2O)_2$ : SCI + $H_2O$ rate constants, $k_5/k_3$, of larger, and more substituted SCI, to be of a similar order of magnitude as for $CH_2OO$ (i.e. 1.5–2.5 × $10^3$)."*

The referee is also right that the dimer reaction will have a negligible impact on the water reaction rates determined in this work because the RH is relatively low and because at the RH where the dimer begins to become a significant loss for *anti*-SCI, almost all of the *anti*-SCI is already being removed by the monomer. Hence there is a negligible effect on the $SO_2$ loss. The paragraph in Section 2.2 now reads:

*"The water dimer reactions of non-$CH_2OO$ SCI are not considered in our analysis. The effect of the water dimer reaction with $C_{10}$ and $C_9$ SCI (rather than the monomer) is expected to be minor at the maximum $[H_2O]$ (2 × $10^{17}$ $cm^{-3}$) used in these experiments (< 30 % RH). Further, with analogy to the syn/anti-$CH_3CHOO$ system, for syn-SCI loss to the dimer (and monomer) will not become competitive at the highest $[H_2O]$ used here; for anti-SCI, the water monomer will already be removing the majority of the SCI at the $[H_2O]$ at which the dimer would become a significant loss process, hence the dimer reaction is deemed unimportant. For $CH_2OO$, the reaction rates with water and the water dimer have been quantified in recent EUPHORE experimental studies, and the values from Newland et al. (2015a) are used in our analysis."*

***In Page 21 lines 23-24, they mention the effective rates for the SCI water vapour reaction at RH 75%, 298 K, and discuss results, but their experimental chamber results are up to RH 28%, so I am not sure it is relevant to mention the results for such high RH.***

We agree with the referee, while the aim of the work is to determine the impact of the SCI under boundary layer conditions, this comment perhaps doesn't belong in the experimental section but instead in a discussion section. We have changed these lines to reflect the experimental conditions from:

*"SCI-3 is expected to undergo unimolecular reactions at least an order of magnitude faster than SCI-4 (Nguyen et al., 2009; Ahrens et al., 2014). The reaction of SCI-3 with water is expected to be slow based on the calculations presented in Table 4, with a pseudo first order reaction rate of 1.0 $s^{-1}$ at 75 % RH, 298 K, whereas the water reaction with SCI-4 is expected to be considerably faster with a pseudo first order reaction rate of 240 $s^{-1}$ at 75 % RH, 298 K. This reaction will thus likely be the dominant fate of SCI-4 at typical atmospheric RH. This is in agreement with the observations of Ma and Marston (2008), that show a clear dependence of nopinone formation on RH*

*(presumed to be formed from SCI + H₂O). Fitting Equation E4 to the data determines values of $\gamma^A$ = 0.41 and $\gamma^B$ = 0.59 (Figure 4)."*

To:

*"SCI-3 is expected to undergo unimolecular reactions at least an order of magnitude faster than SCI-4 (Nguyen et al., 2009; Ahrens et al., 2014). The reaction of SCI-3 with water is expected to be slow based on the calculations presented in Table 4, with a pseudo first order reaction rate of 0.3 $s^{-1}$ at the highest [H₂O] used here, $2 \times 10^{17}$ $cm^{-3}$, 298 K, whereas the water reaction with SCI-4 is expected to be considerably faster with a pseudo first order reaction rate of 85 $s^{-1}$ at [H₂O] = $2 \times 10^{17}$ $cm^{-3}$, 298 K. This reaction would thus be expected to be competitive with reaction with SO₂ for SCI-4 under the experimental conditions employed. This is in agreement with the observations of Ma and Marston (2008), that show a clear dependence of nopinone formation on RH (presumed to be formed from SCI + H₂O). Fitting Equation E4 to the data determines values of $\gamma^A$ = 0.41 and $\gamma^B$ = 0.59 (Figure 4)."*

**Anonymous Referee #2**

***Provide more information how it was done and what´s the accuracy, detection limit etc.***

We have added the following information on instrumental precision to the experimental section:

*"SO₂ and O₃ abundance were measured using conventional fluorescence (reported precision ± 1.0 ppbv) and UV absorption monitors (reported precision ± 4.5 ppbv), respectively;"*

Experimental procedure is detailed clearly in Section 2.1 (p.10, l.23 – p.11, l.14).

***Scavenged the majority of the SCI." Why the authors did not chose perfect experimental conditions for these titration experiments allowing a direct determination of the sCI fraction without any further processing of the primary data?***

It is impossible to scavenge 100 % of the SCI; rates of decomposition of many of the SCI studied are on the order of hundreds per second. There is a limit to how much SO₂ we can safely and practically use in the large EUPHORE chamber (with the lab situated directly below it).

***p.17/18 and table 1: Finally stated sCI yields have a quite low range of uncertainty. Does the uncertainty really reflect the overall precision of this experimental approach?***

We have added a sub-section to Section 2 – Experimental uncertainties which contains the following text:

*"The uncertainty in $k_3/k_2$ was calculated by combining the mean relative errors from the precision associated with the SO₂ and ozone measurements (given in Section 2.1)*

*with the 2σ error and the relative error in φ, using the root of the sum of the squares of these four sources of error. The uncertainty in $k_d/k_2$ was calculated in the same way.*

*The uncertainty in $φ_{min}$ was calculated by combining the uncertainty in $ΔSO_2$ and $ΔO_3$, as above. The uncertainty in φ was calculated by applying the $k_3/k_2$ uncertainties and combining these with the uncertainties in $φ_{min}$, using the root of the sum of the squares."*

**Rate coefficients to set their relative values on an absolute scale, Sheps et al., PCCP (2014). Especially by Taatjes et al., Science (2013). Is there a special reason using the Sheps et al. values? What are the consequences if the Taatjes et al. data are used instead of those by Sheps et al.?**

The difference between the Sheps and the Taatjes measured rate constants for the anti-$CH_3CHOO + SO_2$ reaction is likely owing to differences in the detection techniques used (UV-cavity enhanced absorption spectroscopy vs. Photo-Ionization Mass Spectrometry), with the broadband UV-cavity enhanced absorption technique affording superior sensitivity and selectivity over PIMS (also note that the yield of stabilised anti-$CH_3CHOO$ from the $CH_3CHI + O_2$ reaction is only between 10-30% of the total stabilised $CH_3CHOO$ yield). Therefore, it was decided to put our chamber relative rate measurements on an absolute basis using the Sheps measurements. However, it is important to point out here that our relative rate measurements can be placed on an absolute basis using new and improved evaluated $SO_2$ rate constants as new measurements become available.

**Anonymous Referee #3**

*Scholarly presentation: The text on monoterpene ozonolysis in the early part of the manuscript is a very nice and thorough summary but when read the reader is asking himself: 'And what is the outcome of the present paper for this ?' - this is then treated in the results section. Maybe some on the contents of the introductory text can be shortened and be used when the results are actually presented. That would also compact the paper to some extend. Shortening certain sections and avoiding doubling of text appears advisable as the manuscript reads kind of lengthy at times. There is the danger to loose the reader. The theoretical chemistry section of the paper might be problematic, but I am not an expert in this.*

*Details:*
*Page 4, line 4: The population of CIs is formed...pls check sentence.*

Changed to, "*The population of CIs is formed …*"

*p12, l6: This equation looks strangely formatted. Pls check.*

We're not sure what looks strange about it, but it will in any case be formatted to the ACP style during typesetting.

*p13-l5 - 15: I feel this is partly repeating material already given in the introductory overview. That should be avoided. Please check and discuss the state-of-the art regarding the water reaction, the roles of the water dimer and the difference of syn- and anti-conformers once in the manuscipt and then work with internal referencing.*

We have moved the 'literature review' part of this paragraph to the introduction and included all up to date references. This section now reads:

"*To date, the effects of the water dimer, $(H_2O)_2$ on SCI removal have only been determined experimentally for $CH_2OO$ (Berndt et al., 2014; Chao et al., 2015; Lewis et al., 2015; Newland et al., 2015a; Sheps et al., 2017; Liu et al., 2017) and anti-$CH_3CHOO$ (Lin et al., 2016). Theoretical calculations (Vereecken et al., 2017) have predicted the ratio of the SCI + $(H_2O)_2$ : SCI + $H_2O$ rate constants, $k_5/k_3$, of larger, and more substituted SCI, to be of a similar order of magnitude as for $CH_2OO$ (i.e. 1.5–2.5 × $10^3$).*"

*p16: If it has been shown, that post-CCSD(T) calculations are needed but these cannot be performed for technical reason, what is then the use of this? It is difficult to judge how valid such calculations could be. Certain journals do not accept theoretical chemistry calculation not being performed with the best available techniques. The authors should deal with this. Maybe it is better to outsource this part and do the bigger calculations separately.*

Like experimental measurements, all theoretical predictions are subject to an uncertainty margin, where one aims to reduce the uncertainty by applying the highest possible levels of theory. The methodology used in this work is generally considered high-level and reliable, and the data presented here required well over half a million cpu core hours (>>50 years), and include CCSD(T) calculations with over 1000 basis functions. It is doubtful that any journal would consider these calculations not state-of-the-art for the molecules studied. Going beyond these methodologies is not obvious, and it is not a matter of outsourcing post-CCSD(T) calculations, but rather the question whether anyone is able to do them at all with current computational resources, and can/wants to afford the cost, especially as the empirical corrections described in Vereecken et al. 2017 are expected to recover a large part of the bias on the barrier height. Further improvement could perhaps be made by the kinetic analysis but this, too, requires significant additional computational resources.

As shown in Vereecken et al. 2017, the final rate coefficient predictions for unimolecular reactions are expected to be accurate within a factor of 5. For the CI + $H_2O$ reaction, rate predictions are estimated to be accurate within an order of magnitude. While it would be useful to further reduce the uncertainty on these predictions, they are already sufficiently accurate to have useful predictive value, and the computational cost of reducing the uncertainty may suffer from diminishing returns.

The use of these predictions is that we now have two studies with very different methodologies, experimental and theoretical, which agree quantitatively within the respective uncertainties, suggesting that the conclusions presented in the paper are reliable. Furthermore, the theoretical data allows one to identify the molecular identity of the CI groups used in the experimental analysis; this data is not readily available otherwise.

We have changed the first paragraph of Section 3 to read:

*"The rovibrational characteristics of all conformers of the CI formed from $\alpha$-pinene and $\beta$-pinene, the transition states for their unimolecular reaction, and for their reaction with $H_2O$, were characterized quantum chemically, first using the M06-2X/cc-pVDZ level of theory, and subsequently refined at the M06-2X/aug-cc-pVTZ level. To obtain the most accurate barrier heights for reaction, it has been shown (Berndt et al., 2015; Chhantyal-Pun et al., 2017; Fang et al., 2016a, 2016b; Long et al., 2016; Nguyen et al., 2015) that post-CCSD(T) calculations are necessary. Performing such calculations for the SCI discussed in this paper, with up to 14 non-hydrogen atoms, is well outside our computational resources. Instead, we base our predictions on high-level CCSD(T)/aug-cc-pVTZ single point energy calculations, performed for the reactions of nopinone oxides and the most relevant subset of pinonaldehyde oxides. These data are reliable for relative rate estimates, but it remains useful to further improve the absolute barrier height predictions, as described by Vereecken et al. (2017) based on a data set with a large number of systematic calculations on smaller CI, allowing empirical corrections to estimate the post-CCSD(T) barrier heights. Briefly, they compare rate coefficient calculations against available harmonized experimental and very-high level theoretical*

*kinetic rate predictions, and adjusts the barrier heights by 0.4 to 2.6 kcal mol$^{-1}$ (depending on the base methodology and the reaction type) to obtain best agreement with these benchmark results."*

***p18,l29: Pls check sentence***

Checked.

***p23, Is that section 5.2.4. really needed? I think it should be skipped in order to streamline the whole paper.***

We agree that much of this section is repeated in the conclusions and that parts of the section could be removed to shorten the paper, but also feel that it is useful to provide a summary of the experimental numbers from the previous sections. As such, we have significantly shortened this section, removing the first 15 lines and the final paragraph.

***p24, section 5.3: See general comment on this. Is it necessary to give all the structural data in the SI?***

It is not clear whether the referee is requesting more structural data in the main manuscript, or less in the SI. If it is the latter, then we would suggest that this is exactly what the SI is for. We would hope that the information will be useful for those looking to further understand the theoretical work.

***p 26, Why is section 6 separate from the 'results' section - these are also results, so it might be sensible to make this a sub-point of the results section 5 rather than a new section 6***

We agree that the current grouping of the experimental and theoretical results, but separation of the modelling is somewhat illogical. The three separate techniques: experiment, theory, and modelling are now placed in separate sections. This seems to us a logical and useful way of setting out the paper. In order to clarify the differences between these sections we have renamed Section 5, **Experimental Results**. **Theoretical results and comparison to experiments** is now Section 6. And **Global modeling study** is now Section 7.

***p29, l 14: Oceanic MT emissions are expected to be small compared to the continental ones.***

This may be the case, but we clearly reference two studies which provide values for oceanic MT emissions and then discuss the implication of these studies with reference to our work.

***p30, sections 7 & 8: Maybe these sections can be combined.***

We agree with the referee, Section 7 has been removed and the following sentence has been added to the conclusion (with reference to using a 2-species system for modelling SCI chemistry).

*"Moreover such an approach is required to accurately predict SCI concentrations, which will be underestimated if a simple average of the properties of the two different SCI classes is used."*

**Comment – Rabi Chhantyal-Pun**

*Please could the authors clarify why a value of 819 ($\pm$190) s$^{-1}$ is being used for (CH$_3$)$_2$COO unimolecular reaction rate coefficient? The recent IUPAC task group on atmospheric chemical kinetic data evaluation's preferred value is 397 s$^{-1}$ at 298 K. Is the author's global modelling study affected if more accurate rate coefficients are used?*

The value originally used for the (CH$_3$)$_2$COO decomposition rate (819 s$^{-1}$) comes from ozonolysis experiments (Newland et al., 2015) that used the same experimental conditions as those used in the monoterpene experiments reported in the manuscript. The relative rate from Newland et al. (2015) was scaled to the $k$((CH$_3$)$_2$COO+SO$_2$) rate determined by Huang et al. (2015). This values lies within the uncertainty limits of the recommended IUPAC value.

However, we agree with R. Chhantyal-Pun that the modelling should be done with the IUPAC recommended value (which was not available when the modelling was originally done!). We have repeated the global modeling using the temperature dependent decomposition value from IUPAC ([http://iupac.pole-ether.fr/htdocs/datasheets/pdf/CGI_14_(CH3)2COO+M.pdf](http://iupac.pole-ether.fr/htdocs/datasheets/pdf/CGI_14_(CH3)2COO+M.pdf)) and will include this revised model output in the final manuscript.

As expected, using this slower unimolecular rate increases the concentrations of the SCI-B from ocimene and myrcene, for which the acetone oxide kinetics are used. These increased concentrations lead to an increased relative importance of these SCI compared to other SCI and increased removal of SO$_2$.

The relative contributions of myrcene and ocimene to total [SCI-B] increase from 1.2 % and 5.4 % to 2.7 % and 11% respectively, with commensurate decreases in the relative contributions of the other monoterpenes. Peak annually averaged [SCI] (in the tropics) increases from $1.2 \times 10^4$ cm$^{-3}$ to $1.4 \times 10^4$ cm$^{-3}$.

The contribution of SCI to annual gas phase SO$_2$ oxidation in the terrestrial tropics increases from 1.1 % – 1.2 %. Globally, the annual contribution of SCI to gas phase SO$_2$ oxidation increases from 0.5 % to 0.7 %, and the total annual SO$_2$ removal increases from 6.8 to 8.1 Gg.

All relevant values have been updated throughout the manuscript. The modelling using the Blitz updated k(SO$_2$+OH) rate constant in the SI has also been updated.